# In Situ and Satellite-based Estimates of Cloud Properties and Aerosol-Cloud Interactions over the Southeast Atlantic Ocean

Siddhant Gupta[1, 2*], Greg M. McFarquhar[1, 2], Joseph R. O'Brien[3#], Michael R. Poellot[3], David J. Delene[3], Ian Chang[2], Lan Gao[2], Feng Xu[2], and Jens Redemann[2]

[1]Cooperative Institute for Severe and High-Impact Weather Research and Operations, University of Oklahoma, Norman, OK, USA

[2]School of Meteorology, University of Oklahoma, Norman, OK, USA

[3]Department of Atmospheric Sciences, University of North Dakota, Grand Forks, ND, USA

Now at: *Brookhaven National Laboratory, Upton, NY, USA and #Argonne National Laboratory,
Lemont, IL, USA

Correspondence to: Siddhant Gupta (sgupta@bnl.gov)

**Abstract.** In situ cloud probe data from the NASA ObseRvations of Aerosols above CLouds and their intEractionS (ORACLES) field campaign were used to estimate effective radius ($R_e$), cloud optical thickness ($\tau$), and cloud droplet concentration ($N_c$) for marine stratocumulus over the

southeast Atlantic Ocean. The in situ $R_e$, $\tau$, and $N_c$ were compared with co-located Moderate Resolution Imaging Spectroradiometer (MODIS) retrievals of $R_e$ and $\tau$, and MODIS derived $N_c$. For 145 cloud profiles, a MODIS retrieval was co-located with in situ data with a time gap of less than 1 hour. On average, the MODIS $R_e$ and $\tau$ (11.3 µm and 11.7) were 1.6 µm and 2.3 higher than the in situ $R_e$ and $\tau$ with Pearson's correlation coefficient ($R$) of 0.77 and 0.73, respectively. The

average MODIS $N_c$ (151.5 cm$^{-3}$) was within 1 cm$^{-3}$ of the average in situ $N_c$ with $R$ of 0.90.

        The 145 cloud profiles were classified into 67 contact profiles where aerosol concentration ($N_a$) greater than 500 cm$^{-3}$ was sampled within 100 m above cloud tops and 78 separated profiles where $N_a$ less than 500 cm$^{-3}$ was sampled up to 100 m above cloud tops. Contact profiles had higher in situ $N_c$ (by 88 cm$^{-3}$), higher $\tau$ (by 2.5) and lower in situ $R_e$ (by 2.2



μm) compared to separated profiles. These differences were associated with aerosol-cloud interactions (ACI) and MODIS estimates of the differences were within 5 cm$^{-3}$, 0.5, and 0.2 μm of the in situ estimates when profiles with MODIS $R_e$ > 15 μm or MODIS $\tau$ > 25 were removed. The agreement between MODIS and in situ estimates of changes in $R_e$, $\tau$, and $N_c$ associated with ACI was driven by low biases in MODIS retrievals of cloud properties across different aerosol regimes.

Thus, when combined with estimates of aerosol location and concentration, MODIS retrievals of marine stratocumulus cloud properties over the southeast Atlantic can be used to study ACI over larger domains and longer timescales than possible using in situ data.

## 1 Introduction

Uncertainties in the effective radiative forcing due to aerosol-cloud interactions (ACI) lead

to variability in climate model estimates of Earth's energy budget in future climate scenarios (e.g., Boucher et al., 2013). The ACI for warm, low-level clouds are particularly important due to their dominating impact on the aerosol indirect forcing (Christensen et al., 2016). Further, the shortwave cloud radiative forcing of - 17.1 W m$^{-2}$ (Loeb et al., 2009) is largely driven by the ubiquitous low-level clouds (Hartmann et al., 1992). Marine stratocumulus is the most common

type of low-level cloud with an annual mean coverage of 23 % of Earth's ocean surface (Wood, 2012). The radiative forcing due to well-mixed greenhouse gases (+ 2.83 W m$^{-2}$) (Myhre et al., 2013) or the doubling of $CO_2$ concentration (about + 2.5 W m$^{-2}$) could be offset by the radiative forcing from just a 15 to 20 % reduction in droplet sizes for low clouds (Slingo, 1990). Low-level clouds are thus strong modulators of planetary albedo and global climate.



ACI lead to changes in the cloud radiative forcing through processes that impact cloud

extinction ($\beta$) and optical thickness ($\tau$) which are closely related to microphysical properties like

cloud droplet concentration ($N_c$), effective radius ($R_e$), and liquid water content (LWC). Cloud

radiative forcing is a strong function of $R_e$, which represents the mean droplet size retrieved from

radiative transfer calculations (Hansen and Travis, 1974). An increase in aerosol concentration

($N_a$) can increase the number of cloud condensation nuclei and lead to higher $N_c$ and lower $R_e$

when LWC remains constant. These aerosol-induced changes in $N_c$ and $R_e$ lead to clouds with

higher reflectance or $\tau$ which results in an indirect radiative forcing (Twomey, 1974; 1977).

However, ACI are often masked by meteorological conditions (Mauger and Norris, 2007) and

other cloud responses to increasing $N_a$ like invigoration (Douglas and L'Ecuyer, 2021).

Uncertainties in estimating the impact of ACI on cloud albedo are also driven by differences

between process scales for ACI and the resolution of climate models or satellite retrievals

(McComiskey and Feingold, 2012). The inconsistency in ACI estimates due to the scale differences

is addressed by combining satellite retrievals with airborne observations for specific regimes.

A regime of interest for ACI exists over the southeast Atlantic Ocean where an extensive

stratocumulus deck is overlaid by biomass burning aerosols from southern Africa (Haywood et

al., 2004; Adebiyi and Zuidema, 2016). The biomass burning aerosols exert a direct radiative

forcing by absorbing solar radiation (Cochrane et al., 2019) and heating due to the aerosol

absorption can impact atmospheric stability (Cochrane et al., 2022). Changes in the

thermodynamic profile can lead to changes in cloud properties and result in a semi-direct forcing

(Johnson et al., 2004; McFarquhar and Wang, 2006; Wilcox, 2010). Climate models struggle to

simulate these radiative effects and the altitude of the above-cloud aerosol layer over the



southeast Atlantic leading to biases in model estimates of low-cloud feedbacks and ACI (Das et al., 2020; Mallet et al., 2021).

Airborne campaigns have been conducted over the southeast Atlantic since 2016 to understand the ACI in this region and their impact on global climate (Zuidema et al., 2016; Formenti et al., 2019; Haywood et al., 2021). During the NASA ObseRvations of Aerosols above CLouds and their intEractionS (ORACLES) field campaign (Redemann et al., 2021), in situ measurements of cloud droplet size distributions, from which $N_c$, $R_e$, and $\tau$ can be estimated, were collected over the southeast Atlantic at locations with contact or separation between the

base of the aerosol layer and stratocumulus cloud tops. Variable vertical separation between the aerosol and cloud layers was associated with aerosol-induced changes in $N_c$, $R_e$, and $\tau$ (Gupta et al., 2021, hereafter G21) and precipitation suppression (Gupta et al., 2022, hereafter G22). Satellite retrievals of $N_c$, $R_e$, and $\tau$, and aerosol-induced changes in $N_c$, $R_e$, and $\tau$ could enable such investigations of ACI over larger domains and longer timescales than in situ measurements.

The Earth Observing System Terra and Aqua satellites provide global coverage of cloud microphysical properties using the Moderate Resolution Imaging Spectroradiometer (MODIS). MODIS acquires data for 36 atmospheric bands from 0.4 μm to 14.4 μm including a non-absorbing band (0.86 μm over ocean) which provides information on $\tau$ and a water absorbing band (1.6, 2.1, or 3.7 μm) which provides information on $R_e$ (Platnick et al., 2003). Reflectance

pairs from these bands allow simultaneous retrievals of $R_e$ and $\tau$ (Nakajima and King, 1990). In the absence of direct retrievals, MODIS $N_c$ can be estimated assuming adiabatic LWC (Brenguier et al., 2000; Szczodrak et al., 2001). However, MODIS retrievals of cloud properties have biases



relative to in situ $N_c$, $R_e$, and $\tau$ depending on the cloud type and sampling strategy (Gryspeerdt et al., 2021; Fu et al., 2022), occurrence of drizzle (Zinner et al., 2010), width and shape of droplet

size distributions (Chang and Li, 2002; Brenguier et al., 2011), vertical profile of $R_e$ (McFarquhar and Heymsfield, 1998; Platnick, 2000), and cloud adiabaticity (Min et al., 2012; Merk et al., 2016; Braun et al., 2018). Results from comparisons of MODIS retrievals with in situ data also depend on the cloud probes used for in situ measurements (King et al., 2013; Witte et al., 2018) and the co-location of the MODIS and in situ datasets (Painemal and Zuidema, 2011, hereafter PZ11).

95        A review of satellite-based estimates of $N_c$ concluded that airborne datasets are under-utilized for satellite retrieval evaluation (Grosvenor et al., 2018). This study compares in situ $N_c$, $R_e$, and $\tau$ from ORACLES with MODIS retrievals of $R_e$ and $\tau$ (Platnick et al., 2017a) and the MODIS derived $N_c$ based on the assumption of adiabatic LWC. Previous work comparing MODIS retrievals with in situ observations of marine stratocumulus (PZ11; Min et al., 2012; Noble and Hudson,

2015; Braun et al., 2018; Witte et al., 2018) was extended by using a larger in situ dataset which provides cloud and aerosol measurements under conditions of variable vertical separation between the aerosol and cloud layers. Biases in MODIS retrievals of cloud properties are quantified as a function of the time gap between MODIS retrievals and in situ data. Biases in MODIS Aqua are compared with biases in MODIS Terra and MODIS and in situ estimates of

aerosol-induced changes in $N_c$, $R_e$, and $\tau$ are compared.

The paper is organized as follows. In situ observations and satellite retrievals used in the study are described in Section 2 along with the methodology for spatiotemporal co-location of the in situ and satellite datasets. In Section 3, the MODIS $R_e$, $\tau$, and $N_c$ are compared with in situ



$R_e$, $\tau$, and $N_c$, potential sources of biases are discussed, and uncertainties and errors are

quantified. In Section 4, MODIS estimates of aerosol-induced changes in $R_e$, $\tau$, and $N_c$ over the

southeast Atlantic are compared with in situ estimates. Implications for studies of ACI over the

southeast Atlantic are discussed in Section 5. The conclusions are presented in Section 6.

## 2 Data and Methodology

### 2.1 In situ Observations

115        In situ observations of marine stratocumulus over the southeast Atlantic were collected

during ORACLES using the NASA P-3B aircraft (Redemann et al., 2021). In situ cloud sampling was

conducted during vertical profiles through the stratocumulus layer (hereafter, cloud profiles)

between 10˚ W to 15˚ E and 5˚ N to 20˚ S in September 2016, August 2017, and October 2018

(G22). For each cloud profile, data from in situ cloud probes were used to derive the number

distribution function ($n(D)$) for droplets with diameter ($D$) between 3 to 19200 μm. The cloud

probes used during ORACLES included a Cloud and Aerosol Spectrometer (CAS) (Baumgardner et

al., 2001), three Cloud Droplet Probes (CDPs) (Lance et al., 2010), a Two-Dimensional Stereo

probe (2D-S) (Lawson et al., 2006), a Phase Doppler Interferometer (PDI) (Chuang et al., 2008),

and a High Volume Precipitation Sampler (HVPS-3) (Lawson et al., 1998). A King hot-wire probe

(King et al., 1978) measured LWC (hereafter, King LWC). A Passive Cavity Aerosol Spectrometer

Probe (PCASP) (Cai et al., 2013) measured $n(D)$ for accumulation-mode aerosols ($0.1 < D < 3$ μm).

       The Airborne Data Processing and Analysis processing package (Delene, 2011) was used

to process the CAS, CDP, King hot-wire, and PCASP data. The University of Illinois/Oklahoma

Optical Array Probe Processing Software (UIOOPS) (McFarquhar et al., 2018) was used to process



the 2D-S and HVPS-3 data. A merged droplet size distribution was calculated using the CAS or

CDP dataset for $3 < D < 50$ μm, the 2D-S dataset for $50 < D < 1050$ μm, and the HVPS-3 dataset

for $D > 1050$ μm. $N_c$ was calculated by integrating droplet $n(D)$ from the merged size distribution.

Each 1 Hz data sample with $N_c > 10$ cm$^{-3}$ and King LWC $> 0.05$ g m$^{-3}$ was identified as in-cloud. $N_a$

was calculated by integrating the PCASP $n(D)$ for out of cloud data samples.

135        Due to overlapping measurement ranges, the CAS, the CDPs, and the PDI provided at least

two independent measurements of $n(D)$ for $3 < D < 50$ μm during each flight (G22). Data from

one probe was chosen for inclusion in the merged size distribution based on availability of valid

measurements from the CAS, CDP or PDI and through comparisons between $N_c$ and LWC from

the CAS, CDP, and PDI datasets. The CAS was used to represent droplets with $3 < D < 50$ μm for

research flights from ORACLES 2016 and the CDP for research flights from ORACLES 2017 and

2018 (see G22 for justification and more details). The CAS $n(D)$ for ORACLES 2016 was scaled

using the King LWC for reference due to a potential sizing bias based on comparisons between

the CAS LWC, CDP LWC, and King LWC (G22). The methodology for scaling the 2016 CAS $n(D)$ is

described in Appendix A along with its impact on this study. The uncertainties associated with

the in situ measurements of $N_c$, $R_e$, and $\tau$ are discussed in Appendix B.

For each profile, cloud top height ($Z_T$) and cloud base height ($Z_B$) were defined as the

highest and the lowest altitude, respectively, with $N_c$ and King LWC greater than 10 cm$^{-3}$ and 0.05

g m$^{-3}$, respectively (G21). Cloud thickness ($H$) was defined as the difference between $Z_T$ and $Z_B$. $R_e$

and the effective variance ($V_e$) for the merged size distribution were calculated following Hansen

and Travis (1974) as





$$R_e(h) = \frac{1}{2} \int_3^{19200} D^3 \, n(D,h) \, dD \Big/ \int_0^\infty D^2 \, n(D,h) \, dD$$

and

$$V_e(h) = \int_3^{19200} (D - 2R_e(h))^2 \, D^2 \, n(D,h) \, dD \big/ (2R_e(h))^2 \int_0^\infty D^2 \, n(D,h) \, dD \qquad (1)$$

$R_e$ can also be defined in terms of $R_v$ (mean volume radius) as

$$R_e = k^{-1/3} R_v, \quad k = (1 + d^2)^3 / (ad^3 + 1 + 3 d^2)^2, \qquad (2)$$

where $k$ is the droplet spectral width which is a function of the skewness ($a$) and dispersion ($d$)

of the droplet size distribution (Martin et al., 1994). $k$ can vary depending on aerosol conditions,

occurrence of drizzle, cloud adiabaticity, and height in cloud (McFarquhar and Heymsfield, 2001;

Brenguier et al., 2011). LWC was calculated as

$$LWC(h) = \pi \rho_w / 6 \int_3^{19200} D^3 \, n(D,h) \, dD = 4/3 \, \pi \, \rho_w \, N_c(h) \, R_v(h)^3 \qquad (3)$$

where $h$ is height above $Z_B$ and $\rho_w$ is the liquid water density. At a height $h$ in cloud, LWC is a

function of the average $N_c$ and $R_v$ following Eq. (3). Liquid water path (LWP) and King LWP were

calculated by integrating LWC and King LWC over $h$ from $Z_B$ to $Z_T$. $\tau$ was calculated as

$$\beta_{ext}(h) = \int_3^{19200} Q_{ext} \, \pi/4 \, D^2 \, n(D,h) \, dD, \quad \tau = \int_{Z_B}^{Z_T} \beta_{ext}(h) \, dh, \qquad (4)$$

where $\beta_{ext}$ is the cloud extinction and the extinction efficiency ($Q_{ext}$) for cloud droplets is assumed

to be 2 (Hansen and Travis, 1974) in the limit of geometric optics. The integrals in Eq. (1), (3), and

(4) were converted to discrete sums corresponding to the cloud probe size bins for $D > 3$ μm with

a maximum drop size limit of 19200 μm.



## 2.2 Satellite Retrievals

The MODIS instrument onboard Terra and Aqua acquires passive retrievals of radiance at

non-absorbing and liquid water absorbing spectral bands (Platnick et al., 2003). $R_e$ and $\tau$ are

retrieved using the bispectral retrieval method with the 0.86 μm band paired with the 1.6, 2.1,

or 3.7 μm band (Nakajima and King, 1990). $R_e$ and $\tau$ from the MODIS Collection 6/6.1 Level 2

product (C6) (Platnick et al., 2017a) at 1 km resolution are used. Three retrievals were made for

$R_e$ ($R_{e16}$, $R_{e21}$, and $R_{e37}$) by pairing the 0.86 μm band with the 1.6, 2.1, and 3.7 μm band,

respectively. Consistent with previous studies (e.g., PZ11), $R_{e21}$ was used as the primary retrieval

and MODIS $R_e$ hereafter refers to $R_{e21}$. The wavelength dependence of MODIS $\tau$ is not examined

as $\tau$ is mainly determined by the reflectance from the non-absorbing band (King et al., 1998).

        $R_{e16}$, $R_{e21}$, and $R_{e37}$ represent $R_e$ at 2 to 4 optical depths below cloud top depending on

liquid water absorption and a weighting function based on vertical penetration of photons into

cloud (McFarquhar and Heymsfield, 1998; Platnick, 2000). $R_{e37}$ corresponds to the level closest

to cloud top followed by $R_{e21}$ and $R_{e16}$ in order of increasing distance from cloud top. In an

upgrade from the MODIS Collection 5.1 (C5) product, which reported $R_{e21}$, $R_{e21}$ minus $R_{e16}$, and

$R_{e21}$ minus $R_{e37}$, the MODIS C6 product reported $R_{e16}$, $R_{e21}$, and $R_{e37}$ separately. Thus, biases in $R_{e16}$

and $R_{e37}$ associated with the condition of a successful $R_{e21}$ retrieval are removed (Platnick et al.,

2017b) and $R_{e16}$, $R_{e21}$, and $R_{e37}$ can be compared (Section 3). Within the ORACLES sampling

domain (10˚ W to 15˚ E and 5˚ N to 20˚ S; Fig. 1), $R_{e16}$, $R_{e21}$, and $R_{e37}$ from the C6 product can be

up to 2 μm lower than the corresponding retrievals from the C5 product (Rausch et al., 2017).



The MODIS retrievals are integrated quantities which do not describe the vertical

structure of the cloud. In the absence of in situ data, the vertical profile of LWC and $R_v$ can be

approximated using the adiabatic model to parameterize $N_c$ and LWP as a function of $\tau$ and $R_e$

(Brenguier et al., 2000; Szczodrak et al., 2001). The adiabatic LWC was defined as

$$LWC_{ad}(h) = C_w\,h \; = \; 4/3\,\pi\,\rho_w\,N_{ad}(h)\,R_{vad}(h)^3\,,\tag{5}$$

where $C_w$ is the condensation rate, and the subscript 'ad' represents the adiabatic equivalent of

a variable. Equations (1) to (4) were combined with Eq. (5) to determine $\tau_{ad}$ and LWP$_{ad}$ following

Brenguier et al. (2000) and Szczodrak et al. (2001), respectively, as

$$\tau_{ad} = 3/5\,\pi\,Q_{ext}\,(3\,C_w/4\,\pi\,\rho_w)^{2/3}\,(kN_c)^{1/3}\,H^{5/3}\ \text{and}$$

$$LWP_{ad} = 1/2\,C_w\,H^2\; = 5/9\,\rho_w\,\tau\,R_e\,.\tag{6}$$

Using Equation (5), $N_c$ was parameterized in terms of $\tau$ and $R_e$ following Szczodrak et al.

(2001) as

$$N_c = \sqrt{10}/4\,\pi\,k\;(\alpha\,C_w\,\tau\,/\,\rho_w\,R_e^5\,)^{1/2}\,,\tag{7}$$

where $\alpha$ is the adiabaticity defined as LWP divided by LWP$_{ad}$. MODIS $N_c$ was calculated using

MODIS $R_e$ and $\tau$ based on Eq. (7).

**2.3 Data Selection and Co-location**

MODIS data with valid retrievals within the ORACLES sampling domain (10˚ W to 15˚ E

and 5˚ N to 20˚ S; Fig. 1) were used. The Terra and Aqua satellites overpass the Equator at about

10:30 and 13:30 local time, respectively. Most cloud profiles from ORACLES were flown within 1



to 2 hours of 12:00 UTC. The time gap between the MODIS scan and the in situ sampling for a cloud profile was designated as $\Delta T$. The analysis was limited to cloud profiles with a co-located

MODIS retrieval with $\Delta T$ less than 3600 s. This assumes that the cloud layer did not undergo significant changes within one hour. This assumption was tested by comparing MODIS retrievals against in situ measurements for different upper bounds of $\Delta T$ (Section 3).

MODIS retrievals were co-located with in situ data following the criteria outlined by PZ11. The pixel closest to the cloud top latitude and longitude during a cloud profile was selected. The

location of the selected pixel was adjusted to account for advection of the cloud field using the mean wind speed and direction during the profile from the Turbulent Air Motion Measurement System (Thornhill et al., 2003) on the NASA P-3 aircraft. The wind speed was between 5 to 10 m s$^{-1}$, which meant the pixel location was adjusted by a distance of up to 18 to 36 km over an hour, on average. The MODIS $R_e$ and $\tau$ were averaged over a 5 x 5 pixel domain centered on the

adjusted pixel to account for spatial inhomogeneity. The profile was rejected if the pixel after adjusting for advection was less than 3 pixels from the edge of the MODIS scan and if more than 10 % of the retrievals within the 5 x 5 pixel domain, i.e., at least three out of the 25 pixels, were invalid. Estimates of $Z_T$ and cloud top temperature ($T_T$) from the MODIS C6 product were used to limit the analysis to warm, boundary layer clouds. Four profiles were excluded from the analysis

since the MODIS $Z_T$ was greater than 2500 m or MODIS $T_T$ was less than 273 K.

With the above criteria, at least one cloud profile from 21 research flights conducted during ORACLES had a co-located MODIS retrieval with $\Delta T$ < 3600 s (Table 1). There were 74 cloud profiles with co-located MODIS Terra retrievals and 71 cloud profiles with co-located MODIS



Aqua retrievals (Table 2). $\Delta T$ was evenly distributed with 10 to 15 cloud profiles within every 300

s bin from 0 to 3600 s (except 1200 to 1800 s) (Fig. 2a). For 97 out of the 145 cloud profiles, the

distance between the cloud profile location and the MODIS pixel after adjusting for advection

was below 12 km (Fig. 2b). The distance was greater than 36 km for five profiles.

**3 MODIS versus In situ**

**3.1 $R_e$ Comparisons**

MODIS $R_e$ was compared with the in situ $R_e$ averaged over the top 10 % of the cloud layer

sampled during cloud profiles with a co-located MODIS retrieval with $\Delta T < 3600$ s (Fig. 3a). The

difference between MODIS $R_e$ and in situ $R_e$ for a cloud profile was termed $\Delta R_e$ with positive $\Delta R_e$

indicating that MODIS $R_e$ was greater than in situ $R_e$. The average MODIS $R_e$ (11.3 µm) was greater

than the average in situ $R_e$ (9.7 µm) with Pearson's correlation coefficient ($R$) = 0.77 and root

mean square error (RMSE) = 2.5 µm. All but 12 cloud profiles had positive $\Delta R_e$. The average $\Delta R_e$

was 1.6 ± 1.8 µm where the uncertainty estimate represents the sum of the average retrieval

uncertainty for MODIS $R_e$ from the C6 product and the measurement uncertainty for the average

in situ $R_e$ (Appendix B). Previous comparisons between airborne measurements and MODIS

retrievals of $R_e$ for warm clouds have shown similar $\Delta R_e$ values. For example, the MODIS $R_e$ and

in situ $R_e$ with $\Delta T < 3600$ s for marine stratocumulus over the southeast Pacific had an average

$\Delta R_e$ of 2.1 µm (PZ11). The MODIS $R_e$ and in situ $R_e$ with $\Delta T < 1500$ s for liquid clouds over the

North Atlantic had an average $\Delta R_e$ of 1.7 µm (Painemal et al., 2021).

There were 104 profiles with $\Delta R_e$ less than ± 2 µm while eight profiles had $\Delta R_e > 5$ µm

(Fig. 4a). $\Delta R_e$ was well correlated with MODIS $R_e$ ($R$ = 0.62) and seven out of eight profiles with



$\Delta R_e > 5$ μm had MODIS $R_e > 15$ μm (Fig. 4a). The average $\Delta R_e$ and RMSE decreased from 1.6 to

1.4 and 2.5 to 1.8, respectively, when 13 profiles with MODIS $R_e > 15$ μm were removed. The

MODIS $R_e$ retrieval uncertainty (5 % to 15 %) was poorly correlated with $\Delta R_e$ (Fig. 4b). For lower

bounds of $\Delta T$, the average $\Delta R_e$ and RMSE decreased and the correlation between MODIS $R_e$ and

in situ $R_e$ increased (Table 3). The 42 cloud profiles with a co-located MODIS retrieval with $\Delta T <$

900 s had three profiles with $\Delta R_e > 5$ μm (Fig. 3b). All three of these profiles were associated with

MODIS $R_e > 15$ μm.

MODIS $R_e$ for five out of the eight profiles with $\Delta R_e > 5$ μm was retrieved by MODIS Aqua.

Consequently, retrievals from MODIS Aqua had higher average $\Delta R_e$ and lower correlation with in

situ $R_e$ compared to retrievals from MODIS Terra (Table 3). This was despite the lower average

$\Delta T$ for retrievals from MODIS Aqua (1650 s) compared to retrievals from MODIS Terra (2020 s).

The solar ($\mu_o$) and sensor ($\mu$) zenith angles for MODIS Aqua and MODIS Terra were obtained from

the MODIS C6 product. There were minor differences between the average $\mu_o$ and $\mu$ for MODIS

Terra (24.0˚ and 43.0˚) and MODIS Aqua (29.7˚ and 40.0˚) (Fig. 5). The MODIS $R_e$ and $\Delta R_e$ had

weak correlations with $\mu_o$ ($R = 0.18$ and 0.16) and $\mu$ ($R = -0.05$ and -0.09) which suggests $\mu_o$ and

$\mu$ had little impact on the performance of MODIS Terra relative to MODIS Aqua.

$R_{e16}$, $R_{e21}$, and $R_{e37}$ were compared (Fig. 6) to determine whether $\Delta R_e$ was dependent on

the use of $R_{e21}$ as the primary retrieval. The average $R_{e16}$, $R_{e21}$, and $R_{e37}$ were 10.4, 11.3, and 11.7

μm, respectively. The average $R_{e16}$ and $R_{e21}$ had statistically significant differences while the

average $R_{e21}$ and $R_{e37}$ had statistically insignificant differences. The latter was consistent with

global analyses that found $R_{e37}$ minus $R_{e21}$ depends on cloud regime with positive values (0 to 0.6





μm) for homogeneous marine stratocumulus (Zhang and Platnick, 2011; Fu et al., 2019). Differences between $R_{e16}$, $R_{e21}$, and $R_{e37}$ are associated with differences in the vertical penetration of photons into the cloud. The penetration depth decreases with increasing wavelength from 1.6 to 3.7 μm (Platnick, 2000). An increase in $R_e$ with height in cloud (G22) resulted in $R_{e16} < R_{e21} <$

$R_{e37}$. Although $R_{e21}$ minus $R_{e37}$ depends on $\mu_o$, the average $\mu_o$ for ORACLES (26.8°) was lower than the range of $\mu_o$ (65 to 70°) for which $R_{e37}$ minus $R_{e21}$ exceeds 1 μm (Grosvenor and Wood, 2014). Consistent with Zhang and Platnick (2011), the correlation between $R_{e21}$ and $R_{e16}$ (or $R_{e37}$) decreased for values above 15 μm (Fig. 6). For values below 15 μm, $R_{e16}$, $R_{e21}$, and $R_{e37}$ had an average of 9.9, 10.8, and 11.1 μm, respectively, and high correlation between $R_{e16}$ and $R_{e21}$ ($R$ =

0.92) and $R_{e21}$ and $R_{e37}$ ($R$ = 0.95). Thus, MODIS $R_e$ had a positive bias regardless of the retrieval wavelength. On average, $R_{e21}$ had lower retrieval uncertainty (0.8 μm) compared to $R_{e16}$ (1.9 μm) and $R_{e37}$ (1.1 μm) which suggests $R_{e21}$ gives a robust estimate of the average $\Delta R_e$.

Since each MODIS $R_e$ retrieval penetrates a certain optical depth into cloud, the altitude and in situ $R_e$ at the level of 2 optical depths below cloud top ($Z_{\tau 2}$ and $R_{e\tau 2}$) were compared with

the altitude and in situ $R_e$ averaged over the top 10 % of the cloud ($R_{e10}$ and $Z_{10}$). $R_{e\tau 2}$ and $R_{e10}$ were strongly correlated ($R$ = 0.87) with average values of 9.4 and 9.7 μm, respectively (Fig. 7a). $R_{e\tau 2}$ was less than $R_{e10}$ because the average $Z_{\tau 2}$ was 17 m lower than $Z_{10}$ (Fig. 7b) and $R_e$ increased with height in cloud (G22). When five profiles with $R_e > 15$ μm were removed, $R_{e\tau 2}$ and $R_{e10}$ had average values of 9.3 and 9.4 μm, respectively, with improved correlation ($R$ = 0.95). The average

difference between $R_{e\tau 2}$ and $R_{e10}$ (0.3 μm) was lower than the average $\Delta R_e$ between MODIS $R_e$ and $R_{e10}$ (1.7 μm). Thus, the choice of $R_{e10}$ did not have a large impact on the average $\Delta R_e$.



### 3.2 $\tau$ Comparisons

For profiles with a co-located MODIS retrieval with $\Delta$T < 3600 s, the average MODIS $\tau$ (11.7) was greater than the average in situ $\tau$ (9.4) with $R$ = 0.73 and RMSE = 5.2 (Fig. 8a). $\Delta\tau$ was

defined as the difference between MODIS $\tau$ and in situ $\tau$ for a cloud profile with positive values indicating that MODIS $\tau$ was higher. The average $\Delta\tau$ was 2.3 ± 3.4 where the uncertainty estimate represents the sum of the average retrieval uncertainty for MODIS $\tau$ from the C6 product and the measurement uncertainty for the average in situ $\tau$ (Appendix B). Nine profiles with MODIS $\tau$ > 25 had an average $\Delta\tau$ of 8.1 with six of the profiles having $\Delta\tau$ > ± 10. When profiles with MODIS $\tau$ >

25 were removed, the average $\Delta\tau$ and RMSE decreased from 2.3 to 2.0 and 5.2 to 4.2, respectively. Retrievals from MODIS Terra had lower $\Delta\tau$ and better correlation with in situ $\tau$ compared to retrievals from MODIS Aqua (Table 3). The average $\Delta\tau$ decreased and the correlation between MODIS $\tau$ and in situ $\tau$ improved for profiles with lower $\Delta$T (Table 3). This is consistent with time-dependent improvement in correlations between MODIS $\tau$ and $\tau$ retrieved

using the airborne Solar Spectral Flux Radiometer during ORACLES (Chang et al., 2021).

Profiles with a co-located MODIS retrieval with $\Delta$T < 900 s had $\Delta\tau$ = 1.4, $\sigma(\tau)$ = 2.2, and MODIS $\tau$ uncertainty = 0.6, on average. For 24 out of the 42 profiles with a co-located MODIS retrieval with $\Delta$T < 900 s, $\Delta\tau$ was greater than ± 2 (Fig. 8b). A single profile with $\Delta$T < 900 s had MODIS $\tau$ > 25 which was associated with $\Delta\tau$ of - 14.6. MODIS $\tau$ can have biases relative to in situ

$\tau$ due to spatial heterogeneity of the cloud field or MODIS $\tau$ retrieval uncertainties. On average, MODIS $\tau$ had a standard deviation ($\sigma(\tau)$) of 2.2 over the 25 pixel domain for each cloud profile and $\sigma(\tau)$ was correlated with MODIS $\tau$ ($R$ = 0.72). The $\Delta\tau$ increased with MODIS $\tau$ (Fig. 9a) and the



MODIS $\tau$ retrieval uncertainty increased with MODIS $\tau$ (Fig. 9b). The latter is expected given a decrease in the sensitivity of $\tau$ to the non-absorbing reflectance as $\tau$ increases (King et al., 1998).

However, the average retrieval uncertainty for MODIS $\tau$ (0.6) was less than the average $\Delta\tau$ (2.3).

### 3.3 $N_c$ Comparisons

$N_c$ calculated using MODIS $R_e$ and $\tau$ in Eq. (7) (hereafter, MODIS $N_c$) was compared with in situ $N_c$. Figure 10 shows cloud properties as a function of normalized height above cloud base ($Z_N$) where $Z_N$ equals $Z - Z_B$ divided by $Z_T - Z_B$. The in situ $N_c$ was averaged over the top half of the

cloud layer since entrainment mixing led to lower $N_c$ over the top 10 % of the cloud height (Fig. 10a). Cloud-top entrainment did not affect $R_e$ near cloud top (Fig. 10b) and hence did not impact the comparisons between MODIS and in situ $R_e$. Eight profiles with MODIS $\tau < 5$ were removed from the $N_c$ comparisons to avoid the impact of higher variability in MODIS retrievals for optically thin clouds (Zhang and Platnick, 2011). The exclusion of these profiles did not lead to significant

changes in the $R_e$ or $\tau$ comparisons.

$\Delta N_c$ was defined as the difference between MODIS $N_c$ and in situ $N_c$ for a cloud profile with positive $\Delta N_c$ indicating that MODIS $N_c$ was higher. For 137 profiles with a co-located MODIS retrieval with $\Delta T < 3600$ s and MODIS $\tau > 5$, the average MODIS $N_c$ (151 cm$^{-3}$) had good agreement with the average in situ $N_c$ (151 cm$^{-3}$) with $R = 0.90$ and RMSE = 38 cm$^{-3}$ (Fig. 11). The average $\Delta N_c$

was $0 \pm 64$ cm$^{-3}$ where the uncertainty estimate represents the sum of the error in calculating the average MODIS $N_c$ (Section 3.3.3) and the measurement uncertainty for the average in situ $N_c$ (Appendix B). In comparison, stratocumulus over the southeast Pacific had an average $\Delta N_c$ of - 4 cm$^{-3}$ with $R = 0.94$ (PZ11).


Unlike the $R_e$ or $\tau$ comparisons, lower $\Delta T$ was not associated with lower $\Delta N_c$ or better

correlation between MODIS and in situ $N_c$. Further, MODIS Aqua $N_c$ and MODIS Terra $N_c$ had

similar performance relative to in situ $N_c$ (Table 3). There were 15 profiles with $\Delta N_c$ greater than

± 50 cm$^{-3}$ (average $\Delta N_c$ = 2 cm$^{-3}$ and RMSE = 89 cm$^{-3}$). These profiles were associated with higher

variability in the in situ data with an average standard deviation of 68 cm$^{-3}$ for the in situ $N_c$.

Similarly, the three profiles with $\Delta N_c$ > ± 100 cm$^{-3}$ had an average standard deviation of 86 cm$^{-3}$

for the in situ $N_c$. The correlation between MODIS $N_c$ and in situ $N_c$ increased to 0.93 and the

RMSE decreased to 31 cm$^{-3}$ when these three profiles were removed. For 50 % of the profiles,

$\Delta N_c$ was below ± 20 cm$^{-3}$ which highlights the validity of the adiabatic assumption (Brenguier et

al., 2000; Szczodrak et al., 2001) and the precision of the in situ estimates of $k$, $C_w$, and $\alpha$ (0.76,

2.94 g m$^{-3}$ km$^{-1}$, and 0.74). The agreement between the average MODIS $N_c$ and in situ $N_c$ was

driven by compensating uncertainties associated with the parameters used in Eq. (7). These

uncertainties were examined along with their impact on the calculation of MODIS $N_c$.

### 3.3.1 Uncertainties associated with, $C_w$, $\alpha$, and $k$

MODIS does not retrieve the vertical profile of LWC. Parameters that represent the

estimated rate of condensation with height in cloud ($C_w$) and the ratio of the vertical integrals of

LWC and LWC$_{ad}$ ($\alpha$) can provide the largest sources of error in MODIS $N_c$ (Janssen et al., 2011;

Min et al., 2012). $\alpha$ had a negative correlation with $H$ (Fig. 12) (Min et al., 2012; Braun et al., 2018)

and $C_w$ was a function of cloud base pressure and temperature (Brenguier et al., 2000). Based on

the range of estimates in the existing literature, $C_w$ and $\alpha$ contribute a factor ranging from 0.9 to

1.5 in Eq. (7) (Merk et al., 2016, and references therein). For 142 profiles with a co-located MODIS



retrieval with $\Delta T < 3600$ s and $LWP_{ad} > 5$ g m$^{-2}$, the average $C_w$ and $\alpha$ were $2.94 \pm 0.21$ g m$^{-3}$ km$^{-1}$

and $0.74 \pm 0.26$, respectively, where the uncertainty estimates represent one standard deviation.

These values of $C_w$ and $\alpha$ resulted in a factor of 1.47 in Eq. (7). In comparison, PZ11 assumed $C_w$

$= 2$ g m$^{-3}$ km$^{-1}$ and $\alpha = 1$ with $C_w$ and $\alpha$ contributing a factor of 1.41 in Eq. (7). Using $C_w = 2$ and $\alpha$

$= 1$ in Eq. (7) would decrease MODIS $N_c$ and the average $\Delta N_c$ would change to $- 6$ cm$^{-3}$ (from 0.1

cm$^{-3}$ when $C_w = 2.94$ and $\alpha = 0.74$ were used) while the RMSE remained unchanged.

    $k$ represents spectral width which decreases when droplet size distributions get narrower.

Consistent with PZ11, $k$ averaged over the top 10 % of the cloud layer ($0.76 \pm 0.12$) was higher

than $k$ averaged over the entire cloud layer ($0.70 \pm 0.15$) (Fig. 13), where the uncertainty

estimates represent one standard deviation. Since MODIS $R_e$ corresponds to values near cloud

top, $k = 0.76$ was used in Eq. (7). Using $k = 0.70$ would increase MODIS $N_c$ and the average $\Delta N_c$

and RMSE would change to 13 cm$^{-3}$ and 42 cm$^{-3}$, respectively (from 0 cm$^{-3}$ and 38 cm$^{-3}$ when $k =$

0.76 was used). The value of cloud top $k$ (0.76) was consistent with the $k$ calculated for marine

clouds with entrainment mixing where $k$ decreased when $\alpha$ decreased (Brenguier et al., 2011).

In comparison, higher $k$ (0.8) has been calculated for marine clouds without entrainment mixing

(Martin et al., 1994). The decrease in $N_c$ and LWC near cloud top with increasing $R_e$ was indicative

of inhomogeneous mixing (Fig. 10) and spectral broadening due to entrainment or drizzle (Sinclair

et al., 2021) would explain the higher values for $k$ near cloud top (Fig. 13).

### 3.3.2 Uncertainties associated with MODIS $R_e$ and $\tau$

    The MODIS algorithm assumes vertically homogeneous $R_e$ and LWC (King et al., 1998) but

$R_e$ and LWC increased almost linearly with height (LWC decreased near cloud top due to



entrainment mixing) (Fig. 10b, c). The impact of this inconsistency was examined by quantifying $\Delta N_c$ for profiles with large MODIS biases in $R_e$ or $\tau$. The average $\Delta N_c$ for nine profiles with MODIS $\tau > 25$ (average $\Delta\tau$ = 8.1) and 10 profiles with MODIS $\tau > 5$ and MODIS $R_e > 15$ μm (average $\Delta R_e$ = 4.4 μm) was - 8 and - 15 cm$^{-3}$, respectively. The magnitude of $\Delta N_c$ was greater than 50 cm$^{-3}$ for only two profiles with MODIS $\tau > 25$ and zero profiles with MODIS $R_e > 15$ μm. This suggests a large bias in MODIS $R_e$ or $\tau$ did not necessarily result in a large bias in MODIS $N_c$.

The MODIS algorithm used a modified gamma distribution function to represent the droplet spectrum assuming $V_e$ (Eq. 1) to be 10 % (Platnick et al., 2017b). For such size distributions, $k$ is related to $V_e$ as $k = (1-V_e) \times (1-2V_e)$ and $V_e$ = 10 % corresponds to $k$ = 0.72 (Grosvenor et al., 2018). For ORACLES, $V_e$ decreased with height (Fig. 10d) with a median cloud top $V_e$ of 8.4 % corresponding to $k$ = 0.76. The a priori assumption of $V_e$ = 10 % could lead to biases of up to 1 μm for MODIS $R_e$ (Chang and Li, 2002). Radiative transfer simulations to quantify the MODIS $R_e$ bias associated with $V_e$ were beyond the scope of this study. It is assumed the uncertainties associated with instrument error and atmospheric corrections were included in the retrieval uncertainty estimates in the MODIS C6 product.

The presence of drizzle could introduce biases in MODIS $R_e$ or $N_c$ due to lower $k$ associated with spectral broadening (Sinclair et al., 2021), higher $V_e$ for a bimodal size distribution (Nakajima et al., 2010), or lower $\alpha$ due to cloud water removal through precipitation (Braun et al., 2018). However, the average rain rate for ORACLES was too low (0.06 mm h$^{-1}$) (G22) for drizzle to have a major impact on the $R_e$ retrievals (Zinner et al., 2010; PZ11). This was supported by the positive values for $R_{e37}$ minus $R_{e21}$ which represent size distributions without a significant drizzle mode



(Nakajima et al., 2010). The impact of cloud water removal through precipitation was included by using the in situ $\alpha$ (0.74) in Eq. (7).

### 3.3.3 MODIS $N_c$ Error Analysis

The total error for MODIS $N_c$ from Eq. (7) was quantified using the propagation of measurement uncertainties associated with $k$, $C_w$, and $\alpha$, and retrieval uncertainties associated with MODIS $R_e$ and $\tau$. Assuming the covariances were normally distributed and random, the total error can be calculated using Gaussian error propagation as

$$\left(\frac{\delta N_c}{N_c}\right)^2 = \left(\frac{1}{2}\frac{\delta \tau}{\tau}\right)^2 + \left(\frac{5}{2}\frac{\delta R_e}{R_e}\right)^2 + \left(\frac{1}{2}\frac{\delta C_w}{C_w}\right)^2 + \left(\frac{1}{2}\frac{\delta \alpha}{\alpha}\right)^2 + \left(\frac{\delta k}{k}\right)^2, \tag{8}$$

where $\delta$ represents the error for each variable. For MODIS $R_e$ and $\tau$, the error was defined as the average retrieval uncertainty from the MODIS C6 product (7.5 and 5 %, respectively). For $k$, $C_w$, and $\alpha$, the error was defined as one standard deviation (16, 7.1, and 35 % of their averages).

     Based on Eq. (8), MODIS $N_c$ had an error of 30.5 %. This was smaller than previous estimates of 38 % (Janssen et al., 2011) and 78 % (Grosvenor et al., 2018). Consistent with 410 Grosvenor et al. (2018), $R_e$ was the parameter with the largest contribution to the total error in MODIS $N_c$ followed by $\alpha$ and $k$. Profiles with MODIS $R_e$ > 15 μm and average $\Delta R_e$ of 4.4 μm had an average $\Delta N_c$ of - 15 cm$^{-3}$ highlighting the compensation of the $R_e$ uncertainty Eq. (7) by the other parameters. MODIS $N_c$ calculated using in situ estimates of $k$, $C_w$, and $\alpha$ from ORACLES was higher than the MODIS $N_c$ determined using a priori assumptions for $k$, $C_w$, and $\alpha$. For example, 415 substituting $C_w$ = 2 g m$^{-3}$ km$^{-1}$ and $\alpha$ = 1 (PZ11) and $k$ = 0.8 (Martin et al., 1994) in Eq. (7) would introduce a factor which was 9 % lower than using $C_w$ = 2.94 g m$^{-3}$ km$^{-1}$, $\alpha$ = 0.74, and $k$ = 0.76.



The MODIS $N_c$ calculated based on these a priori assumptions would have average $\Delta N_c$ and RMSE of -14 cm$^{-3}$ and 39 cm$^{-3}$, respectively (compared to 0 cm$^{-3}$ and 38 cm$^{-3}$ using the in situ estimates).

## 4 Aerosol-cloud Interactions

During the ORACLES research flights, variable vertical separation was observed between biomass burning aerosols from southern Africa and marine stratocumulus over the southeast Atlantic (Redemann et al., 2021). Cloud profiles were conducted at locations of both contact and separation between the base of the aerosol layer and the top of the cloud layer. Cloud profiles with aerosol concentration ($N_a$) greater than 500 cm$^{-3}$ within 100 m above cloud tops were

termed "contact profiles" and cloud profiles with $N_a < 500$ cm$^{-3}$ up to 100 m above cloud tops were termed "separated profiles" (G21).

Across the ORACLES campaigns, 173 contact profiles were conducted with 84 to 90 cm$^{-3}$ higher in situ $N_c$, 1.4 to 1.6 μm lower in situ $R_e$, and 0.04 to 3.06 higher in situ $\tau$ compared to 156 separated profiles (G22). These differences between in situ $N_c$, $R_e$, and $\tau$ for contact and

separated profiles were statistically significant ($p < 0.02$) and their ranges represent the 95 % confidence intervals from a two-sample t-test. These confidence intervals represent the difference between the average values for contact and separated profiles determined with 95 % confidence. Given the statistically similar sea surface temperature, lower tropospheric stability, and estimated inversion strength at the locations of contact and separated profiles, the cloud

microphysical differences were attributed to aerosol-cloud interactions (G22).

A co-located MODIS retrieval with $\Delta T$ less than 3600 s was available for 67 contact and 78 separated profiles (Table 1). These contact profiles had 84 to 91 cm$^{-3}$ higher in situ $N_c$, 1.4 to 1.6



µm lower in situ $R_e$, and 0.44 to 4.64 higher in situ $\tau$ compared to the separated profiles. When the in situ $N_c$ and $R_e$ were averaged over the top 50 % and top 10 % of the cloud, respectively,

contact profiles had 87 to 98 cm$^{-3}$ higher in situ $N_c$ and 1.5 to 2.1 µm lower in situ $R_e$ compared to separated profiles. Differences between the in situ $N_c$, $R_e$, and $\tau$ for contact and separated profiles were compared with corresponding differences between MODIS $N_c$, $R_e$, and $\tau$. For simplicity, it is assumed the MODIS and in situ uncertainties were consistent for contact and separated profiles. This assumption allows direct comparison of MODIS estimates of the differences between cloud properties for contact and separated profiles with in situ estimates.

differences between cloud properties for contact and separated profiles with in situ estimates.

For contact profiles, the average MODIS $R_e$ (9.9 µm) was 1.4 µm greater than the average in situ $R_e$ with $R$ = 0.76 (Fig. 14). In comparison, for separated profiles, the average MODIS $R_e$ (12.6 µm) was 1.9 µm larger than the average in situ $R_e$ with $R$ = 0.72. Separated profiles had a greater $\Delta R_e$ compared to contact profiles because 12 out of 13 profiles with MODIS $R_e$ > 15 µm,

with high average $\Delta R_e$ (4.0 µm) (Fig. 4a), were classified as separated profiles. As a result, the MODIS $R_e$ estimate (2.6 µm) for the aerosol-induced increase in $R_e$ from contact to separated profiles was greater than the in situ $R_e$ estimate (2.1 µm). MODIS $R_e$ had a similar positive bias for contact and separated profiles with MODIS $R_e$ < 15 µm (1.3 and 1.5 µm, respectively). Thus, when profiles with MODIS $R_e$ > 15 µm were removed, the estimate of the $R_e$ difference between contact

and separated profiles using MODIS $R_e$ and in situ $R_e$ were closer (1.8 µm and 1.6 µm, respectively). Fewer profiles with $R_e$ from MODIS Terra had MODIS $R_e$ > 15 µm compared to MODIS Aqua and closer agreement was observed between the in situ $R_e$ and MODIS $R_e$ estimates of the aerosol-induced change in $R_e$ for MODIS Terra compared to MODIS Aqua (Table 4).



For contact profiles, the average MODIS $\tau$ (13.3) was 2.5 optical depths greater than the

average in situ $\tau$ with $R = 0.75$ (Fig. 15). In comparison, for separated profiles, the average MODIS

$\tau$ (10.3) was 2.1 optical depths greater than the average in situ $\tau$ with $R = 0.62$. As a result, the

MODIS $\tau$ estimate (3.0) was greater than the in situ $\tau$ estimate (2.6) of the aerosol-induced

increase in $\tau$ from separated to contact profiles. Contact profiles with co-located MODIS Aqua

retrievals had lower in situ $\tau$ compared to separated profiles. The $\tau$ from MODIS Aqua reproduced

the sign and magnitude of this change (Table 4). The MODIS Terra $\tau$ underestimated the in situ

$\tau$ increase from separated to contact profiles (Table 4) due to the profile with MODIS $\tau > 25$ and

$\Delta\tau = -14.6$ (Fig. 15). All nine profiles with MODIS $\tau > 25$ were classified as contact profiles (Fig.

15). When these profiles were removed given their large average $\Delta\tau$ (8.1), the remaining 58

contact profiles had MODIS $\tau$ (10.8) which was 1.6 optical depths greater than in situ $\tau$ ($R = 0.74$),

on average. Subsequently, the MODIS $\tau$ estimate (0.5) was less than the in situ $\tau$ estimate (1.0)

of the aerosol-induced increase in $\tau$ from separated to contact profiles.

The average MODIS $N_c$ for contact profiles (203 cm$^{-3}$) was 2 cm$^{-3}$ lower than the average

in situ $N_c$ ($R = 0.86$) (Fig. 16). For separated profiles, the average MODIS $N_c$ (105 cm$^{-3}$) was 2 cm$^{-3}$

greater than the average in situ $N_c$ ($R = 0.82$). This meant the estimate for the aerosol-induced

increase in $N_c$ (from separated to contact profiles) from MODIS $N_c$ (99 cm$^{-3}$) was similar to the

estimate from in situ $N_c$ (103 cm$^{-3}$). The three profiles with $\Delta N_c > \pm 100$ cm$^{-3}$ were classified as

contact profiles. When these profiles were removed, estimates for the aerosol-induced increase

in $N_c$ from separated to contact profiles from MODIS $N_c$ and in situ $N_c$ were similar (95 cm$^{-3}$ and

94 cm$^{-3}$, respectively). For MODIS Terra retrievals, underestimation of the increase in in situ $N_c$



from separated to contact profiles (Table 4) was driven by the profile with $\Delta\tau$ = - 14.6 and MODIS

$\tau$ > 25 (Fig. 15). When this profile was removed, the MODIS $N_c$ and in situ $N_c$ estimates were

within 5 cm$^{-3}$. The MODIS $N_c$ calculated using a priori assumptions for $k$, $C_w$, and $\alpha$ underestimated

the in situ $N_c$ for contact profiles (by 20 cm$^{-3}$) and separated profiles (by 8 cm$^{-3}$). The a priori

MODIS $N_c$ estimate (91 cm$^{-3}$) for the increase in $N_c$ from separated to contact profiles was slightly

lower than the in situ $N_c$ estimate (103 cm$^{-3}$).

**5 Discussion**

Differences between climate model and observational estimates of the effective radiative

forcing due to ACI are largely driven by uncertainties in observational estimates of the radiative

forcing due to aerosol effects on cloud albedo (RF$_{aci}$) (Gryspeerdt et al., 2020). Issues with satellite

estimates of RF$_{aci}$ persist due to biases in satellite retrievals of $N_c$ (Grosvenor et al., 2018), above-

cloud aerosol properties (Meyer et al., 2015; Painemal et al., 2020; Chang et al., 2021), and

aerosol perturbations of $N_c$ (Quaas et al., 2020). Factors that frequently result in biases in MODIS

retrievals of cloud properties include subpixel heterogeneity (Zhang and Platnick, 2011), solar

and satellite viewing geometry (Grosvenor and Wood, 2014; Painemal et al., 2021), cloud

thermodynamic phase (Ahn et al., 2018), and drizzle occurrence (Zinner et al., 2010; Sinclair et

al., 2021). These factors had limited impact on MODIS retrievals used in this study due to the low

latitude of the ORACLES domain and observations of homogeneous, warm, closed cell marine

stratocumulus over the southeast Atlantic with low precipitation rates (G21; G22).

Satellite estimates of $N_c$ and aerosol perturbations of $N_c$ over the southeast Atlantic have

biases within 10 % of the in situ estimates. The differences between the MODIS and in situ $R_e$ or



$\tau$ were reduced by screening data with MODIS $R_e$ > 15 μm or MODIS $\tau$ > 25, respectively. This is consistent with the improvement in correlations between MODIS $N_c$ and in situ $N_c$ from multiple field campaigns when using a threshold of maximum $R_e$ of around 15 μm (Gryspeerdt et al., 2021). The MODIS-based screening led to MODIS estimates of aerosol-induced changes in $N_c$, $R_e$, and $\tau$ within 5 cm$^{-3}$, 0.5 μm, and 0.7 of the in situ estimates. Agreement between the MODIS and in situ estimates of aerosol-induced changes in $N_c$, $R_e$, and $\tau$ was associated with consistent biases in MODIS retrievals of cloud properties across different aerosol regimes. Such agreement suggests cloud properties for horizontally homogeneous, warm, closed cell marine stratocumulus can be estimated using MODIS retrievals in the absence of in situ datasets.

Better accuracy in remote sensing retrievals of the aerosol layer is needed to constrain the uncertainties in satellite estimates of RF$_{aci}$ over the southeast Atlantic (Douglas and L'Ecuyer, 2020). In particular, biases in satellite estimates of the placement or optical and microphysical properties of the above-cloud aerosol layer need to be addressed (Rajapakshe et al., 2017; Painemal et al., 2020; Peers et al., 2021). The High Spectral Resolution Lidar Generation 2 (HSRL-2) (Hair et al., 2008) was used to measure aerosol extinction and backscatter at 355, 532, and 1064 nm during all three ORACLES campaigns. Research is ongoing to use HSRL-2 data for estimating the vertical profile of cloud condensation nuclei (Lenhardt, 2021). Accounting for the attenuation of upwelling solar radiation by above-cloud absorbing aerosols over the southeast Atlantic could increase the average MODIS $\tau$ and $R_e$ by up to 9 % and 2 %, respectively (Meyer et al., 2015). The Research Scanning Polarimeter (RSP) (Cairns et al., 1999) was used during ORACLES to collect polarimetric retrievals of cloud properties (Alexandrov et al., 2012) which do not operate under the assumptions required for MODIS retrievals. RSP retrievals can help examine





biases in MODIS retrievals of clouds with higher precipitation rates or bimodal size distributions (Sinclair et al., 2021; Fu et al., 2022) or complicated solar and viewing geometry (e.g., Painemal et al., 2021). Future work will use RSP retrievals combined with other airborne datasets to evaluate MODIS retrievals while accounting for above-cloud aerosols (e.g., Chang et al., 2021).

**6 Conclusions**

In situ measurements of $N_c$, $R_e$, and $\tau$ for marine stratocumulus over the southeast Atlantic were collected during the NASA ORACLES field campaign. In situ data from 145 cloud profiles were co-located with MODIS retrievals from the Terra and Aqua satellites with $\Delta$T less than 1 hour. The average MODIS $R_e$ and $\tau$ (11.3 μm and 11.7) were greater than the average in situ $R_e$ and $\tau$ (9.7 μm and 9.4) with $R$ = 0.77 and 0.73, respectively. The average bias in MODIS $R_e$ was 1.6 ± 1.8 μm and the average bias in MODIS $\tau$ was 2.3 ± 3.4, where the uncertainty represents the sum of the average MODIS retrieval uncertainty and the in situ measurement uncertainty. MODIS $N_c$ (151 cm$^{-3}$) had an estimated calculation error of 30.5 % and showed good agreement with in situ $N_c$ (151 cm$^{-3}$) with $R$ = 0.90 and an average bias of 0 ± 64 cm$^{-3}$. The retrieval uncertainty for MODIS $R_e$ provided the largest source of error in calculating MODIS $N_c$ but compensating uncertainties for $\tau$, $k$, $C_w$, and $\alpha$ resulted in good agreement. Cloud profiles with an $N_c$ bias greater than 50 cm$^{-3}$ were associated with higher variability in the in situ $N_c$. The biases in MODIS $R_e$ and $\tau$ were lower for lower bounds of $\Delta$T and for retrievals from MODIS Terra compared to MODIS Aqua. Profiles with MODIS $R_e$ > 15 μm had larger biases in MODIS $R_e$ (average bias = 4.5 μm) and profiles with MODIS $\tau$ > 25 had larger biases in MODIS $\tau$ (average bias = 8.1).





Variability in the vertical profile of absorbing aerosols over the southeast Atlantic was associated with changes in $N_c$, $R_e$, and $\tau$ under similar meteorological conditions. There were 67 545 "contact" profiles where $N_a > 500$ cm$^{-3}$ was sampled within 100 m above cloud tops while 78 "separated" profiles had $N_a < 500$ cm$^{-3}$ up to 100 m above cloud tops. Contact profiles had higher in situ $N_c$ and $\tau$ (88 cm$^{-3}$ and 2.5 higher) and lower in situ $R_e$ (2.2 $\mu$m lower) compared to separated profiles. MODIS retrievals were able to estimate the sign of these aerosol-induced changes in $N_c$, $R_e$, and $\tau$. The magnitude of the MODIS estimates of differences between contact and separated 550 profiles was within 5 cm$^{-3}$, 0.5, and 0.2 $\mu$m of the in situ estimates when profiles with MODIS $R_e$ > 15 $\mu$m or MODIS $\tau$ > 25 were removed.

The agreement between MODIS and in situ estimates of aerosol-induced changes in cloud microphysical properties over the southeast Atlantic was associated with similar biases in MODIS retrievals across different aerosol conditions. This motivates the use of MODIS retrievals to study 555 ACI for homogeneous marine stratocumulus over a larger domain of the southeast Atlantic and over longer timescales than is possible using in situ data. Future work will be aimed at improving lidar and polarimetric retrievals of the vertical profile and microphysical and optical properties of absorbing aerosols over the southeast Atlantic layers and the underlying cloud properties (Zeng et al., 2014; Rajapakshe et al., 2017; Painemal et al., 2020; Lenhardt, 2021).

**APPENDIX A – Scaling the CAS/CDP *n(D)* based on King LWC**

For ORACLES 2016, CAS data were used in the analysis since CDP measurements were invalid due to an instrument misalignment issue. G22 showed there were statistically significant differences between the average CAS LWC of 0.15 ± 0.09 g m$^{-3}$ (± one standard deviation) and





the average King LWC of 0.28 ± 0.15 g m$^{-3}$ ($R$ = 0.80). The LWC comparison provides an estimate

of the uncertainties in the CAS data due to known issues like coincidence of particles in the

sample volume (Lance et al., 2012) and uncertainties in the collection geometry (e.g.,

Baumgardner et al., 2017). Comparisons between CAS and CDP $N_c$ (when CDP data were

available) indicate the CAS may be affected by coincidence of particles within the sample volume.

However, accounting for coincidence while processing the CAS data affected $N_c$ by less than 2 %.

Based on a recommendation by the manufacturers of CAS (Droplet Measurement Technologies,

DMT), a sample area of 0.26 mm$^2$ was used to process CAS droplet counts to obtain $N_c$ instead of

using 0.13 mm$^2$ from the CAS manual.

For the six flights selected for analysis, the King LWC and CAS LWC had a best fit slope ($a$)

between 0.46 and 0.63 and $R$ = 0.71 to 0.93 (Table A1). Therefore, an adjustment is used to

increase the CAS LWC to match King LWC. The simplest way to do this would be to increase the

CAS sample area, which is a first order adjustment that assumes the CAS is sizing the droplets

correctly. However, based on the LWC differences, it is hypothesized the CAS was under-sizing

the droplets passing through the CAS sample volume. The methodology outlined by PZ11 was

thus used to account for the sizing bias wherein the CAS $n(D)$ was scaled by adjusting the CAS size

bins using the King LWC as a reference by setting

$$CAS\ LWC = a\ x\ King\ LWC\ . \tag{A1}$$

The scaled midpoint diameter for the i$^{th}$ CAS size bin ($D_i^*$) is determined as

$$D_i^* = a^{-1/3}\ D_i\ , \tag{A2}$$



where $D_i$ is the midpoint diameter for the $i^{th}$ CAS size bin. The $D_i$ used to calculate the CAS $R_e$ and

LWC is replaced by $D_i^*$ to calculate the scaled CAS $R_e$ and LWC. The CAS size bin midpoints were

thus increased (by up to 30 %) since $D_i^* > D_i$ for $a < 1$ and each flight had $a < 1$. The average in situ

$R_e$ for the 34 profiles from ORACLES 2016 with a co-located MODIS retrieval (Table 2) increased

from 8.6 μm for unscaled CAS $n(D)$ to 10.6 μm for CAS $n(D)$ scaled using Eqs. (A1) and (A2).

The average MODIS $R_e$ (12.4 μm) overestimated the average in situ $R_e$ from both the

unscaled and scaled CAS $n(D)$. When the CAS $n(D)$ was scaled, the number of profiles having in

situ $R_e$ > MODIS $R_e$ increased from 0 to 2 and the average $\Delta R_e$ decreased from 3.8 μm ($R = 0.83$)

to 1.8 μm ($R = 0.86$), relative to using the unscaled CAS $n(D)$. These changes were consistent with

the hypothesis of CAS under sizing the droplets passing through the CAS sample volume. Since

the average $\Delta R_e$ for scaled CAS $n(D)$ was consistent with previous studies (PZ11; Painemal et al.,

2021), the scaled CAS $n(D)$ was used in the analysis.

Valid CDP measurements were available for ORACLES 2017 and 2018. For the research

flights from ORACLES 2017 and 2018, the average CDP LWC was 0.18 ± 0.16 g m⁻³ and 0.21 ± 0.14

g m⁻³, the average King LWC was 0.21 ± 0.15 g m⁻³ and 0.20 ± 0.12 g m⁻³, and the average CAS

LWC was 0.09 ± 0.07 g m⁻³ and 0.10 ± 0.07 g m⁻³, respectively (G22). The differences between the

King LWC and the CDP LWC are within the typical uncertainties of these in situ cloud probes

(Baumgardner et al., 2017). Nevertheless, the impact of scaling the CDP data was investigated

using Eqs. (A1) and (A2) to determine if this would lead to qualitative changes in the results.

For 14 out of 18 flights from ORACLES 2017 and 2018, the King LWC and CDP LWC had 0.7

< $a$ < 1.4 and the CDP size bin midpoints were adjusted by less than 13 % following Eq. (A2). When



the CDP $n(D)$ was scaled for the 42 profiles from ORACLES 2017, the average CDP $R_e$ increased

from 7.6 μm to 8.7 μm, the number of profiles having in situ $R_e$ > MODIS $R_e$ increased from 2 to

21, and the average $\Delta R_e$ decreased from 1.4 μm ($R$ = 0.57) to 0.3 μm ($R$ = 0.43), relative to using

the unscaled CDP $n(D)$. Scaling the CDP $n(D)$ led to a decrease in the best fit slope for MODIS $R_e$

as a function of in situ $R_e$ (0.73 to 0.50) along with an increase in the intercept (3.5 to 4.7 μm).

These changes suggest the in situ $R_e$ might be overestimated when the CDP $n(D)$ is scaled, and

the unscaled CDP $n(D)$ was thus used in the study for ORACLES 2017. Given this and the closer

agreement between CDP LWC and King LWC (compared to CAS LWC and King LWC), it is unlikely

the CDP had a sizing bias like the CAS and thus the unscaled CDP $n(D)$ was used in the analysis.

When the CDP $n(D)$ was scaled for the 73 profiles from ORACLES 2018, the average CDP

$R_e$ increased from 10.5 μm to 10.8 μm, the number of profiles having in situ $R_e$ > MODIS $R_e$

increased from 9 to 15, and the average $\Delta R_e$ decreased from 1.9 μm ($R$ = 0.68) to 1.6 μm ($R$ =

0.62), relative to using the unscaled CDP $n(D)$. The use of scaled CDP $n(D)$ led to small changes in

the best fit slope for MODIS $R_e$ as a function of in situ $R_e$ (0.77 to 0.73) and the intercept (4.3 to

4.5 μm). Scaling the CDP $n(D)$ for ORACLES 2018 did not have a major impact on the CDP dataset.

To remain consistent with the use of unscaled CDP data for ORACLES 2017, unscaled CDP data

were used in the study for ORACLES 2018, as well.

When MODIS $R_e$ was compared with in situ $R_e$ calculated using unscaled $n(D)$ for all three

campaigns, the average $\Delta R_e$ was 2.2 μm with $R$ = 0.72 and a best-fit slope and intercept of 0.86

and 3.5 μm, respectively (Fig. A1a). In comparison, when MODIS $R_e$ was compared with in situ $R_e$

calculated using scaled $n(D)$ for all three campaigns, the average $\Delta R_e$ was 1.3 μm with $R$ = 0.70





and a best-fit slope and intercept of 0.90 and 2.4 µm, respectively (Fig. A1b). The use of either

scaled or unscaled $n(D)$ for all three campaigns did not lead to qualitative changes in the results

presented in the study. MODIS $R_e$ always had a positive bias greater than 1 µm relative to in situ

$R_e$. It must be noted that the quantitative changes highlight the uncertainties associated with in

situ data which must be considered when validating satellite retrievals using airborne datasets

(Witte et al., 2018).

**Appendix B - In situ Measurement Uncertainties**

The error for in situ measurements of $N_c$, $R_e$, and $\tau$ depend on droplet sizing and

concentration uncertainties associated with limitations of instrument measurement principles

and data processing algorithms (Baumgardner et al., 2017; McFarquhar et al., 2017). Although

sources of in situ measurement uncertainty are relatively well known, there is no established

methodology for calculating sizing and concentration uncertainties or propagating uncertainties

to the error for in situ $N_c$, $R_e$, or $\tau$. A single probe is unable to characterize the entire spectrum of

cloud droplets, and droplet size distributions are derived by combining number distribution

function from scattering and imaging probes (G22). This complicates uncertainty estimation and

error propagation for in situ measurements. After accounting for instrument and data processing

uncertainties, droplet sizing and concentration uncertainties can be ± 20 % and ± 50 % for imaging

probes and ± 50 % and ± 20 % for scattering probes (Baumgardner et al., 2017).

Three approaches are examined for estimating the error for in situ $N_c$, $R_e$, and $\tau$. First,

sizing and concentration uncertainties of 10 % each are assumed throughout the size distribution

(Baumgardner et al., 2017) to derive a minimum estimate of the error. Second, uncertainties are





estimated based on inter-comparisons between cloud probes with similar measurement size ranges. Third, the standard error of the mean, defined as the standard deviation divided by the square root of the sample size, is calculated. For each variable, the maximum estimate out of the three approaches is designated as the error estimate.

For the first approach, the droplet concentration ($N_c$) uncertainty is 10 %. Sizing and concentration uncertainties are not always independent, and Gaussian error propagation can underestimate the error. Thus, error ($\delta$) in $R_e$ and $\tau$ is determined using the maxima and minima concentration and size as

$$\delta x = \frac{x(D+\delta D, N(D)+\delta N(D)) - x(D-\delta D, N(D)-\delta N(D))}{2} \quad , \quad x = \{\tau, R_e\} \tag{B1}$$

where $\delta D$ = 0.1 $D$ and $\delta N(D)$ = 0.1 $N(D)$.

Following Eq. (3), $\delta \tau$ equals 0.3 $\tau$ and $\delta R_e$ equals 0.1 $R_e$. The fractional estimate for $\delta \tau$ is greater than the equivalent estimate from Gaussian error propagation (0.22 $\tau$) while the estimate for $\delta R_e$ is equivalent to the Gaussian error estimate. Following this approach, the average in situ $N_c$, $R_e$, and $\tau$ error estimates are 15 cm$^{-3}$, 1.0 μm, and 2.8, respectively. For the second approach, average values of $N_c$, $R_e$, and $\tau$ from the scaled CAS datasets (Appendix A) are compared with the PDI dataset for ORACLES 2016 and with the CDP datasets for ORACLES 2017 and 2018 based on data availability (G22). Across deployments, the relative difference between $N_c$, $R_e$, and $\tau$ from the cloud probes was within 12.5 %, 10 %, and 21 %, respectively. Thus, the average in situ $N_c$, $R_e$, and $\tau$ error estimates are 19 cm$^{-3}$, 1.0 μm, and 2.0, respectively. For the third approach, the standard deviation is divided by the square root of the sample size to determine the standard error of the mean. The $N_c$, $R_e$, and $\tau$ error estimates are 7.4 cm$^{-3}$ , 0.2 μm, and 0.5, respectively.



Using the highest error estimate out of the three approaches, the average in situ $N_c$, $R_e$, and $\tau$ along with the error estimate are 150 ± 19 cm$^{-3}$, 9.7 ± 1.0 μm, and 9.4 ± 2.8, respectively.

Uncertainty estimates for biases in MODIS retrievals relative to in situ measurements (Section 3) are defined as the sum of the retrieval uncertainty and calculation error for MODIS $N_c$, $R_e$ and $\tau$, and the in situ measurement uncertainty. The average MODIS $N_c$ was 150 ± 45 cm$^{-3}$ and the bias in MODIS $N_c$ was 0 ± 64 cm$^{-3}$. The average MODIS $R_e$ was 11.3 ± 0.8 μm and the bias in MODIS $R_e$ was 1.6 ± 1.8 μm. The average MODIS $\tau$ was 11.7 ± 0.6 and the bias in MODIS $\tau$ was 2.3 ± 3.4. The

average biases in MODIS retrievals relative to in situ measurements were within the MODIS retrieval and in situ measurement uncertainty for all three variables.

*Code availability*. University of Illinois/Oklahoma Optical Array Probe (OAP) Processing Software is available at https://doi.org/10.5281/zenodo.1285969 (McFarquhar et al., 2018). The Airborne Data Processing and Analysis software package is available at

https://doi.org/10.5281/zenodo.3733448 (Delene et al., 2020).

*Data availability*. All ORACLES data are accessible via digital object identifiers (DOIs) under the references: https://doi.org/10.5067/Suborbital/ORACLES/P3/2018_V2 (ORACLES Science Team, 2020a), https://doi.org/10.5067/Suborbital/ORACLES/P3/2017_V2 (ORACLES Science Team, 2020b), https://doi.org/10.5067/Suborbital/ORACLES/P3/2016_V2 (ORACLES Science Team,

2020c). The MODIS Collection 6 Cloud product is available at dx.doi.org/10.5067/MODIS/MOD06_L2.061 (Platnick et al., 2017a, last access: May 26, 2022).

*Author contributions.* SG designed the study and analyzed the data with guidance from GMM and inputs from IYC, LG, FX, and JR. JRO'B, DJD, and MRP processed the cloud probe and PCASP data.



SG processed 2D-S and HVPS-3 data. All authors were involved with the ORACLES field campaign.

GMM, MRP, and JR acquired funding. SG wrote the manuscript with guidance from GMM and

reviews from all co-authors.

*Competing interests*. The authors declare that they have no conflicts of interest.

*Special issue statement*. This article is part of the special issue "New observations and related

modeling studies of the aerosol–cloud–climate system in the Southeast Atlantic and southern

Africa regions (ACP/AMT inter-journal SI)". It is not associated with a conference.

*Acknowledgements.* We acknowledge the entire ORACLES science team for their contributions

during data acquisition and analysis. We thank the NASA Ames Earth Science Project Office and

the NASA P-3B crew for logistical and aircraft support. Some of the computing for this project

was performed at the OU Supercomputing Center for Education & Research (OSCER) at the

University of Oklahoma (OU).

*Financial support.* Funding for this project was obtained from NASA Award #80NSSC18K0222.

ORACLES is funded by NASA Earth Venture Suborbital-2 grant NNH13ZDA001N-EVS2. SG was

supported by NASA headquarters under the NASA Earth and Space Science Fellowship grants

NNX15AF93G and NNX16A018H and by 80NSSC18K0222.




Table 1: List of research flights analyzed and the time range, number, sampling duration (in parentheses), and cloud top height ($Z_T$) for profiles with a co-located MODIS retrieval with time
gap ($\Delta T$) less than 3600 s. Number and duration listed for profiles classified by above-cloud aerosol location.

| Flight Date | Time (UTC) | Separated | Contact | $Z_T$ (m) |
|---|---|---|---|---|
| 06 Sep 2016 | 09:36 – 12:35 | 6 (256 s) | 9 (606 s) | 509 - 1002 |
| 10 Sep 2016 | 10:08 – 12:36 | 5 (255 s) | 0 (0 s) | 1151 - 1201 |
| 14 Sep 2016 | 09:36 – 13:02 | 3 (148 s) | 0 (0 s) | 635 - 814 |
| 20 Sep 2016 | 12:57 – 13:11 | 0 (0 s) | 2 (61 s) | 580 - 583 |
| 25 Sep 2016 | 11:00 – 13:51 | 6 (363 s) | 3 (148 s) | 729 - 1124 |
| 12 Aug 2017 | 11:53 – 13:46 | 0 (0 s) | 8 (327 s) | 1148 - 1193 |
| 13 Aug 2017 | 10:15 – 11:33 | 0 (0 s) | 15 (718 s) | 1334 - 1384 |
| 15 Aug 2017 | 12:55 – 13:27 | 0 (0 s) | 6 (169 s) | 1108 - 1148 |
| 21 Aug 2017 | 13:34 – 13:35 | 1 (18 s) | 0 (0 s) | 1447 |
| 24 Aug 2017 | 12:39 – 12:40 | 0 (0 s) | 1 (10 s) | 1099 |
| 28 Aug 2017 | 11:46 – 13:18 | 4 (168 s) | 7 (496 s) | 1070 - 1230 |
| 27 Sep 2018 | 10:07 – 13:11 | 10 (366 s) | 0 (0 s) | 819 - 1169 |
| 30 Sep 2018 | 09:50 – 12:24 | 6 (183 s) | 7 (337 s) | 747 - 840 |
| 03 Oct 2018 | 13:29 – 13:30 | 1 (13 s) | 0 (0 s) | 1157 |
| 07 Oct 2018 | 11:03 – 11:14 | 0 (0 s) | 3 (136 s) | 845 - 928 |
| 10 Oct 2018 | 10:16 – 13:31 | 2 (153 s) | 1 (42 s) | 991 - 1329 |
| 12 Oct 2018 | 13:12 – 14:19 | 3 (61 s) | 0 (0 s) | 1431 - 1905 |
| 15 Oct 2018 | 10:28 – 13:09 | 4 (125 s) | 0 (0 s) | 693 - 1547 |
| 19 Oct 2018 | 12:36 – 13:00 | 9 (661 s) | 0 (0 s) | 959 - 1276 |
| 21 Oct 2018 | 10:21 – 12.25 | 10 (504 s) | 0 (0 s) | 675 - 812 |
| 23 Oct 2018 | 10:28 – 13:08 | 8 (286 s) | 5 (317 s) | 873 - 1281 |
| **Total (2016)** | | **20 (1,022 s)** | **14 (815 s)** | |
| **Total (2017)** | | **5 (186 s)** | **37 (1,720 s)** | |
| **Total (2018)** | | **53 (2,352 s)** | **16 (832 s)** | |
| **Total** | | **78 (3,560 s)** | **67 (3,367 s)** | |

Table 2: Number of cloud profiles during ORACLES deployments with a co-located MODIS Terra
or Aqua retrieval for $\Delta T$ less than 3600, 1800, or 900 s.

| $\Delta T$ | Terra (2016, 2017, 2018) | Aqua (2016, 2017, 2018) | Total |
|---|---|---|---|
| 3600 s | 20, 15, 39 | 14, 27, 30 | 145 |
| 1800 s | 9, 3, 17 | 12, 13, 12 | 66 |
| 900 s | 9, 1, 10 | 8, 7, 7 | 42 |



Table 3: Average bias ($\Delta$), root mean square error (RMSE), and Pearson's correlation coefficient ($R$) for MODIS (Terra, Aqua, combined) retrievals relative to in situ $R_e$, $\tau$, and $N_c$ for different $\Delta$T.

| Parameter | $\Delta$T (s) | Terra $\Delta$, RMSE ($R$) | Aqua $\Delta$, RMSE ($R$) | Combined $\Delta$, RMSE ($R$) |
|---|---|---|---|---|
| | 3600 | 1.5, 2.1 (0.82) | 1.8, 2.9 (0.75) | 1.6, 2.5 (0.77) |
| $R_e$ (μm) | 1800 | 1.4, 1.5 (0.95) | 2.1, 3.2 (0.78) | 1.8, 2.6 (0.81) |
| | 900 | 1.3, 1.5 (0.91) | 1.8, 2.8 (0.81) | 1.6, 2.3 (0.83) |
| | 3600 | 2.8, 6.1 (0.70) | 1.9, 4.2 (0.73) | 2.3, 5.2 (0.73) |
| $\tau$ | 1800 | 1.7, 5.0 (0.90) | 1.8, 4.0 (0.72) | 1.8, 4.5 (0.85) |
| | 900 | 1.3, 5.1 (0.91) | 1.6, 4.5 (0.51) | 1.4, 4.8 (0.86) |
| | 3600 | 0, 42 (0.87) | -1, 32 (0.93) | 0, 38 (0.90) |
| $N_c$ (cm$^{-3}$) | 1800 | 11, 53 (0.82) | 4, 32 (0.95) | 7, 43 (0.90) |
| | 900 | 9, 57 (0.74) | 10, 34 (0.96) | 10, 46 (0.87) |

Table 4: Differences between the average $R_e$, $\tau$, and $N_c$ for contact and separated profiles based on MODIS retrievals (Terra, Aqua, and combined) and in situ measurements. Positive values indicate contact profiles had a higher value.

| Parameter | $\Delta$T (s) | Terra (In situ) | Aqua (In situ) | Combined (In situ) |
|---|---|---|---|---|
| | 3600 | -1.7 (-1.4) | -3.6 (-2.9) | -2.6 (-2.1) |
| $R_e$ (μm) | 1800 | -0.9 (-0.7) | -5.6 (-3.5) | -3.4 (-2.2) |
| | 900 | -0.3 (-0.4) | -5.9 (-3.5) | -3.1 (-2.0) |
| | 3600 | 6.0 (6.1) | -0.8 (-1.5) | 3.0 (2.6) |
| $\tau$ | 1800 | 7.1 (10.1) | -0.0 (-1.1) | 2.4 (3.0) |
| | 900 | 7.3 (10.5) | -2.6 (-3.1) | 1.4 (2.6) |
| | 3600 | 83 (87) | 115 (118) | 99 (103) |
| $N_c$ (cm$^{-3}$) | 1800 | 80 (91) | 153 (139) | 113 (111) |
| | 900 | 43 (77) | 159 (131) | 99 (102) |

Table A1: ORACLES 2016 flight dates with the best fit slope ($a$) and intercept ($c$) between the
average CAS LWC and King LWC from the flight.

| Flight date | $a + c$ ($R$) |
|---|---|
| September 06 | 0.51 + 0.01 (0.71) |
| September 10 | 0.63 - 0.02 (0.93) |
| September 12 | 0.47 + 0.00 (0.88) |
| September 14 | 0.55 - 0.04 (0.85) |
| September 20 | 0.60 + 0.01 (0.88) |
| September 25 | 0.46 + 0.04 (0.74) |





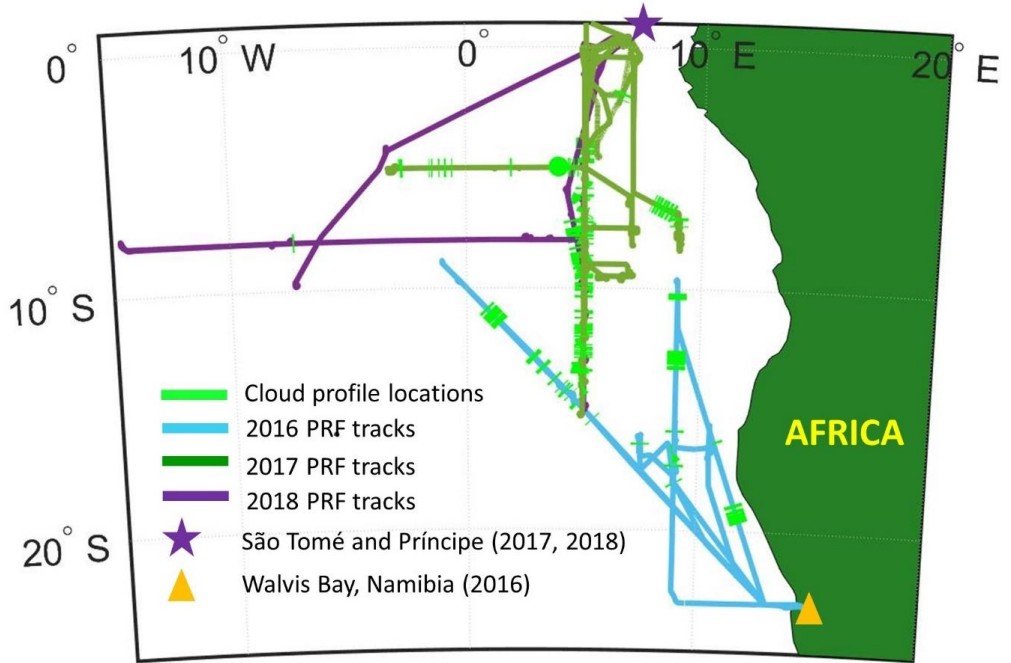

Figure 1: ORACLES flight tracks, base of operations, and sampling locations for profiles with a MODIS retrieval co-located with in situ data for ΔT less than 3600 s.

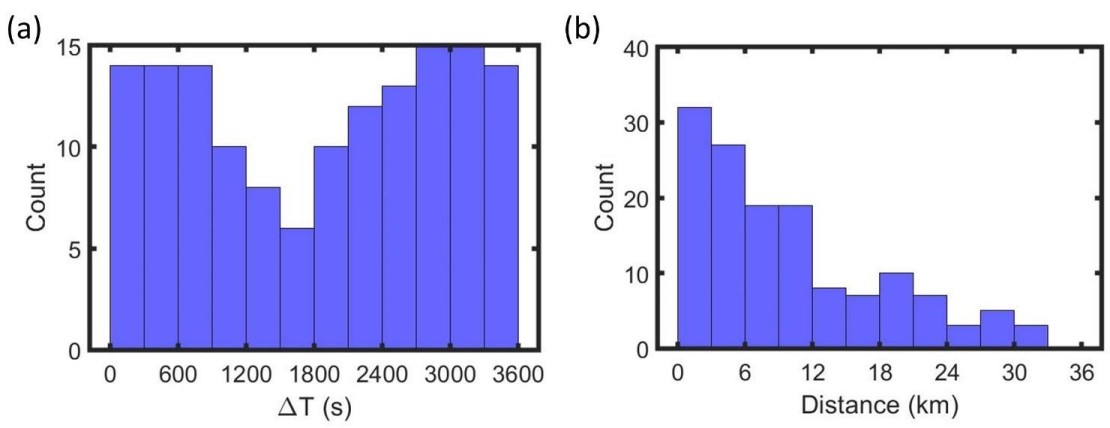


Figure 2: Histograms of (a) time gap between profiles and the co-located MODIS scan (ΔT) and (b) distance between profiles and the co-located MODIS pixel after adjusting for advection.



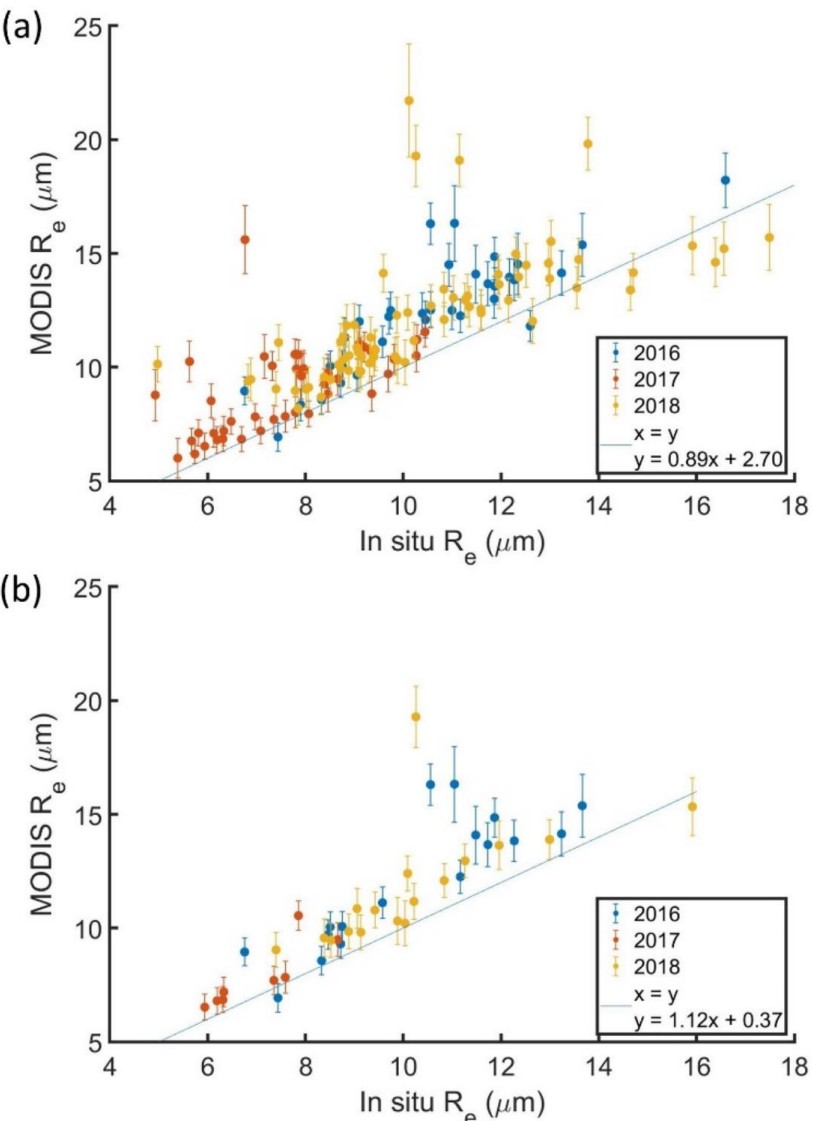

Figure 3: MODIS $R_e$ versus in situ $R_e$ for profiles with a MODIS retrieval co-located with in situ
data for $\Delta T$ (a) less than 3600 s and (b) less than 900 s colored by ORACLES deployment year.
Each point represents a cloud profile with the in situ $R_e$ averaged over the top 10 % of the cloud
and MODIS $R_e$ averaged over a 5 x 5 pixel domain.





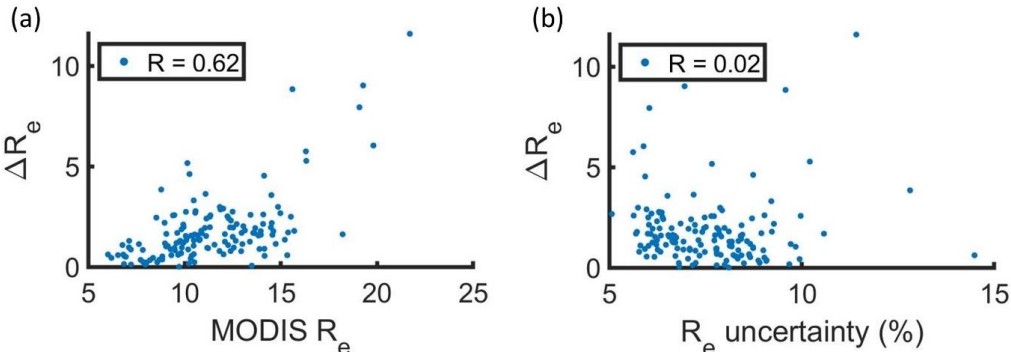

Figure 4: Magnitude of the difference between MODIS $R_e$ and in situ $R_e$ ($\Delta R_e$) for profiles with a
MODIS retrieval co-located with in situ data for $\Delta T$ less than 3600 s as a function of (a) MODIS
$R_e$ and (b) MODIS $R_e$ uncertainty. Each point represents the average over a 5 x 5 pixel domain.

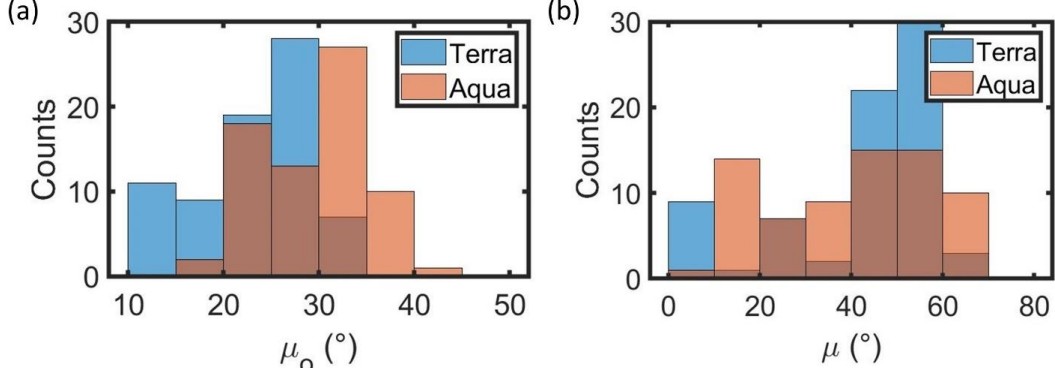

Figure 5: Histograms of (a) solar zenith angle ($\mu_o$) and (b) sensor zenith angle ($\mu$) for MODIS
retrievals co-located with in situ data for $\Delta T$ less than 3600 s.

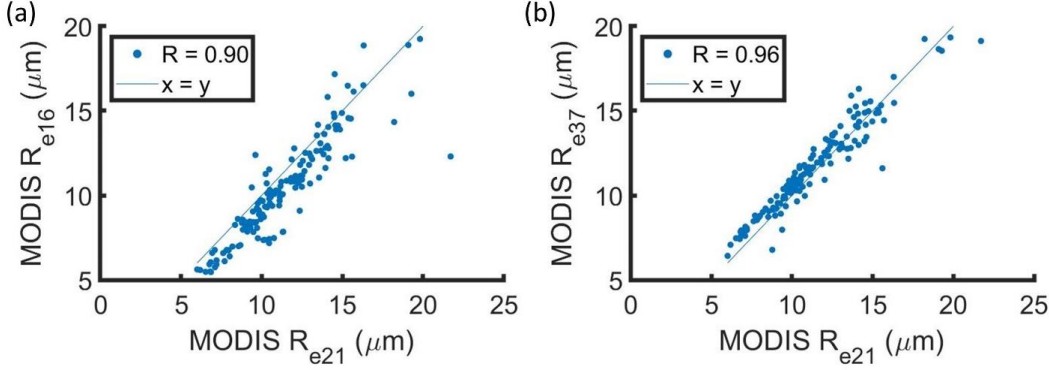

Figure 6: (a) $R_{e16}$ and (b) $R_{e37}$ as a function of $R_{e21}$ for MODIS retrievals co-located with in situ
data for $\Delta T$ less than 3600 s. Each point represents average values over a 5 x 5 pixel domain.





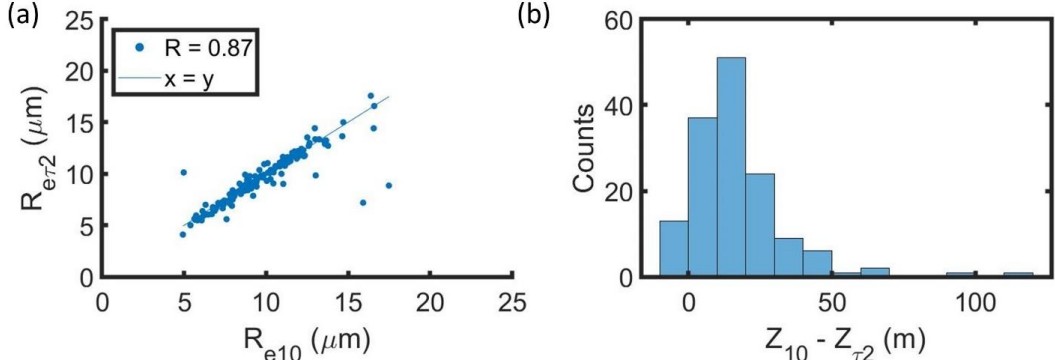

Figure 7: (a) Scatter between $R_e$ at two optical depths below cloud top ($R_{e\tau2}$) versus $R_e$ averaged
over top 10 % of cloud layer ($R_{e10}$) and (b) histogram of the difference between $Z_{10}$ and $Z_{\tau2}$ for
profiles with a MODIS retrieval co-located with in situ data for $\Delta T$ less than 3600 s.



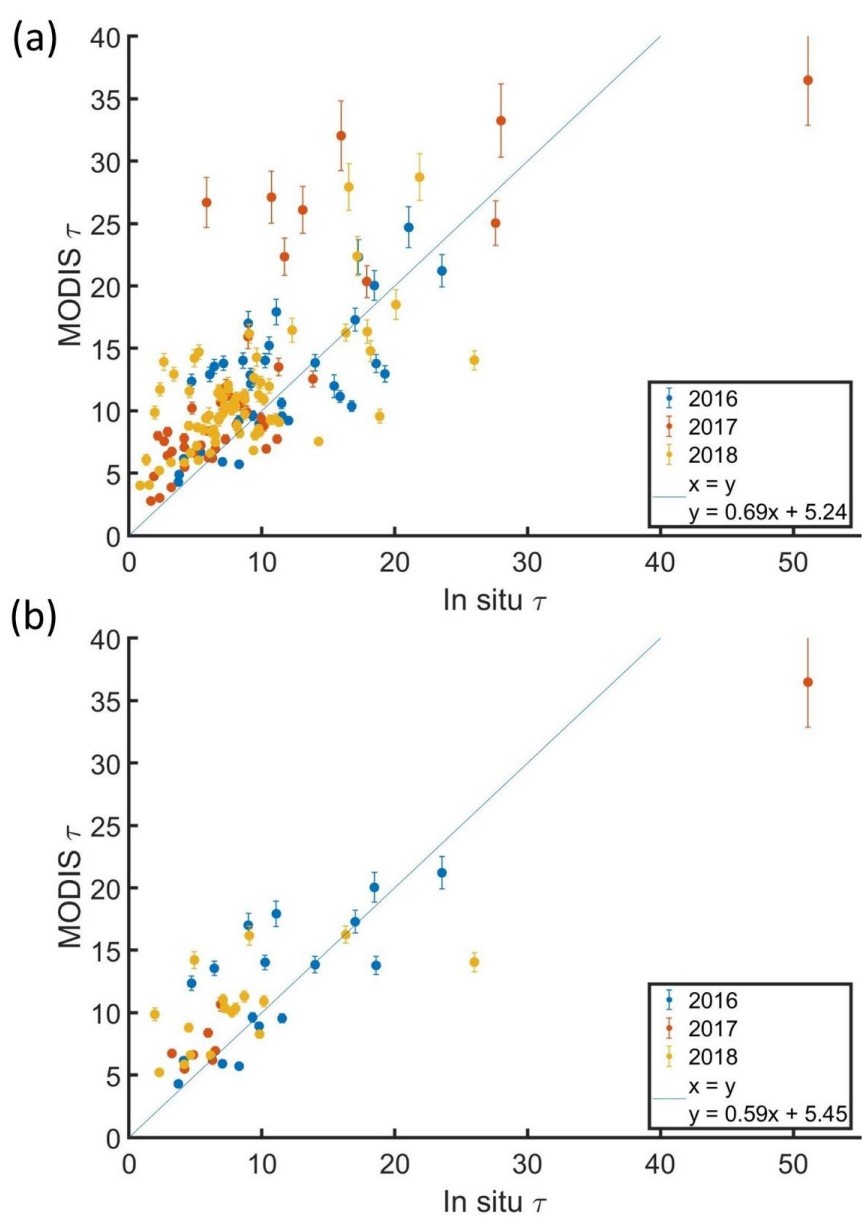

Figure 8: MODIS $\tau$ versus in situ $\tau$ for profiles with a MODIS retrieval co-located with in situ data for $\Delta T$ (a) less than 3600 s and (b) less than 900 s colored by ORACLES deployment year. Each point represents a cloud profile with the MODIS $\tau$ averaged over a 5 x 5 pixel domain.


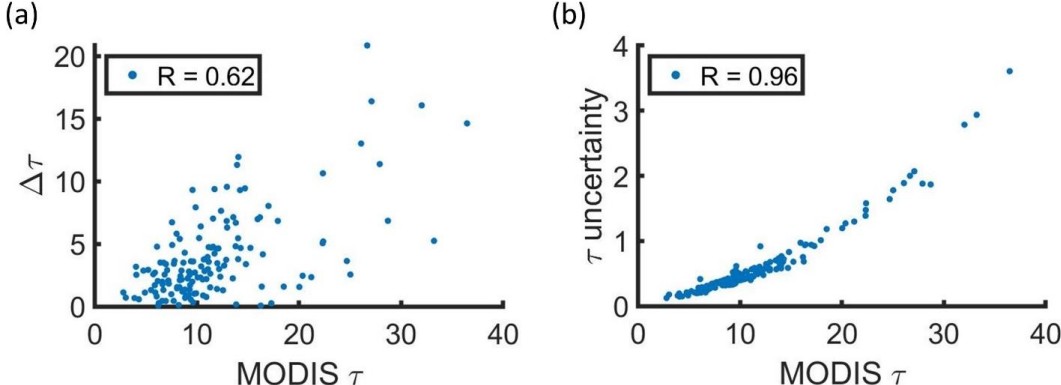

Figure 9: MODIS $\tau$ versus (a) magnitude of the difference between MODIS $\tau$ and in situ $\tau$ ($\Delta$) and (b) MODIS $\tau$ retrieval uncertainty for profiles with a MODIS retrieval co-located with in situ data for $\Delta$T less than 3600 s. Each point represents average values over a 5 x 5 pixel domain.

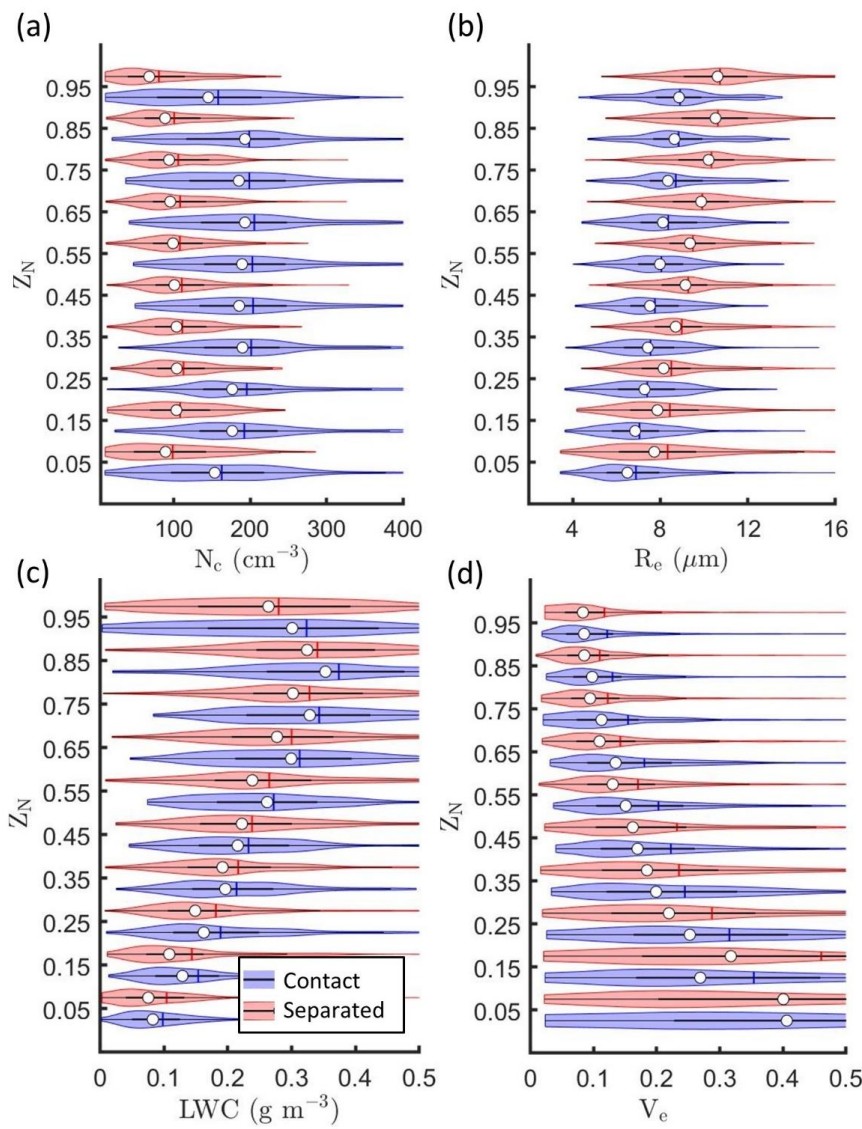


Figure 10: Kernel density estimates (indicated by width of shaded area) and boxplots showing mean (vertical line) and median (white circle) for (a) $N_c$, (b) $R_e$, (c) LWC, and (d) $V_e$ versus normalized height in cloud ($Z_N$) for profiles with a MODIS retrieval co-located with in situ data for $\Delta$T less than 3600 s.





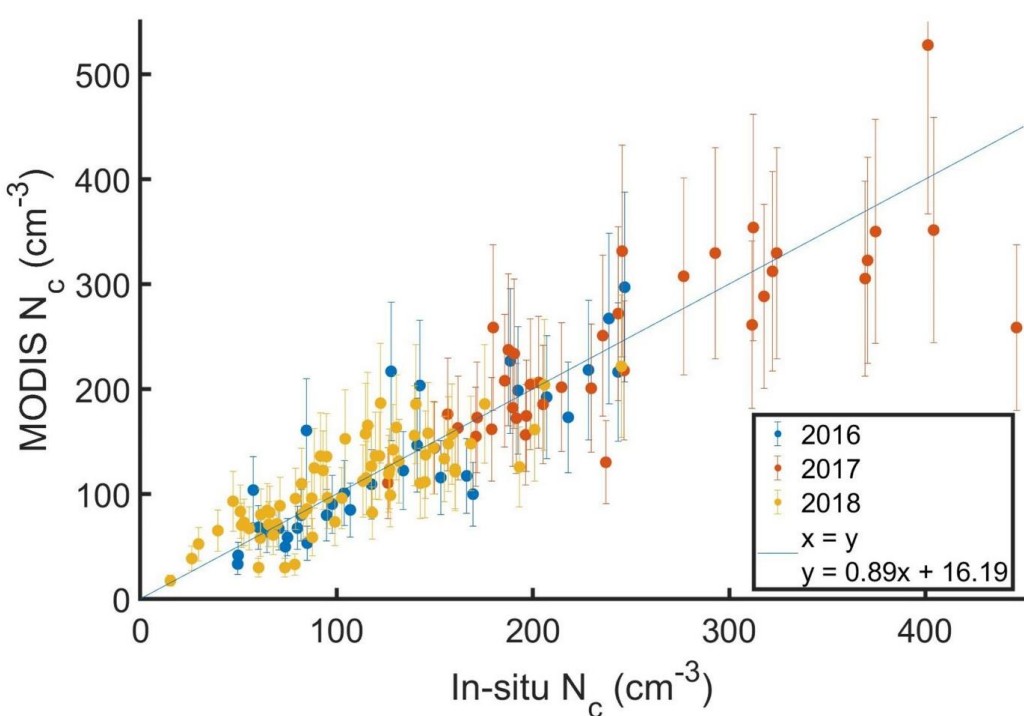


Figure 11: MODIS $N_c$ versus in situ $N_c$ for with a MODIS retrieval co-located with in situ data for $\Delta T$ less than 3600 s colored by ORACLES deployment year. Each point represents a cloud profile with the in situ $N_c$ averaged over the top half of the cloud and MODIS $N_c$ calculated using MODIS $R_e$ and $\tau$ averaged over a 5 x 5 pixel domain.

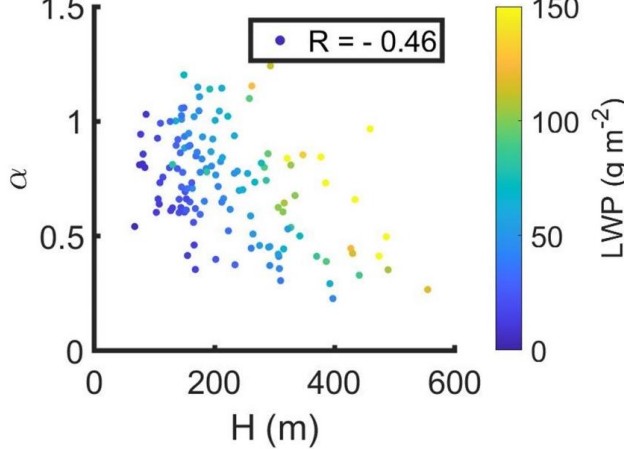


Figure 12: Cloud adiabaticity ($\alpha$) versus cloud thickness ($H$) colored by liquid water path (LWP) for with a MODIS retrieval co-located with in situ data for $\Delta T$ less than 3600 s.



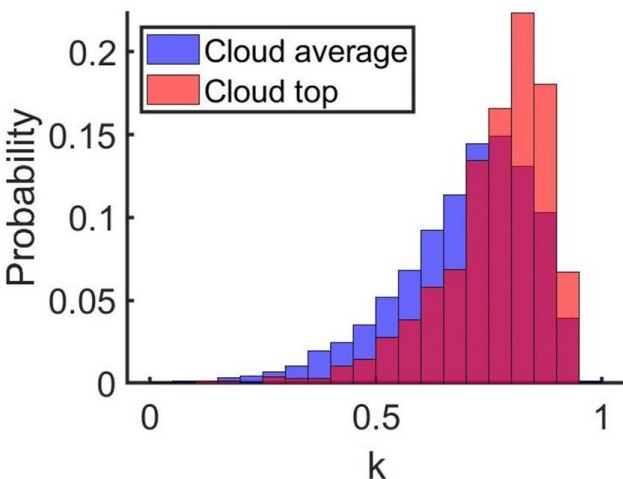

Figure 13: Probability density function for *k* averaged over entire cloud layer (blue) or top 10 %
of cloud (red) for profiles with a MODIS retrieval co-located with in situ data for ΔT less than
3600 s.

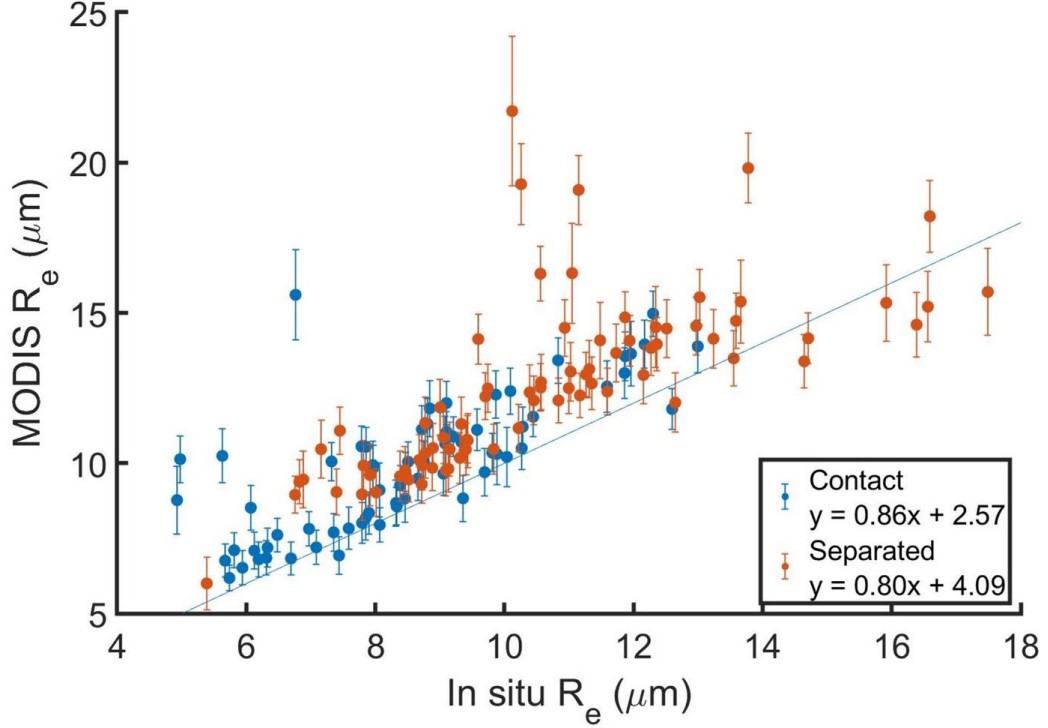

Figure 14: Same as Fig. 3a with cloud profiles colored based on regime classification.



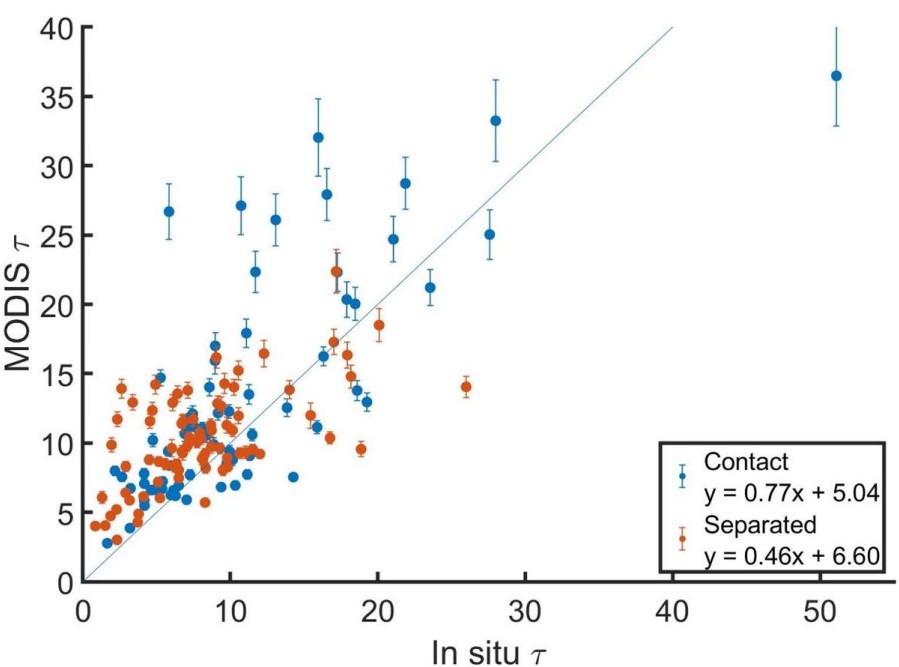

Figure 15: Same as Fig. 8a with cloud profiles colored based on regime classification.

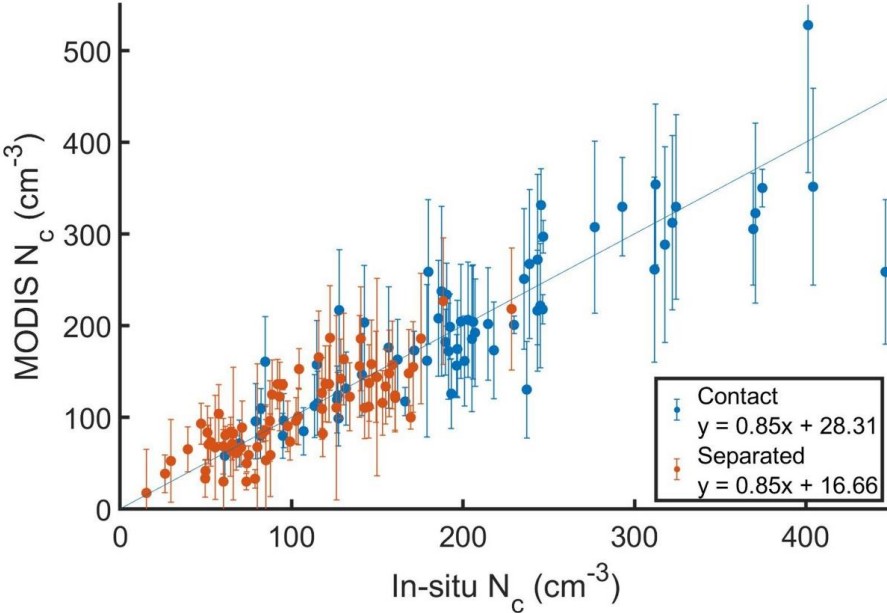

Figure 16: Same as Fig. 11 with cloud profiles colored based on regime classification.

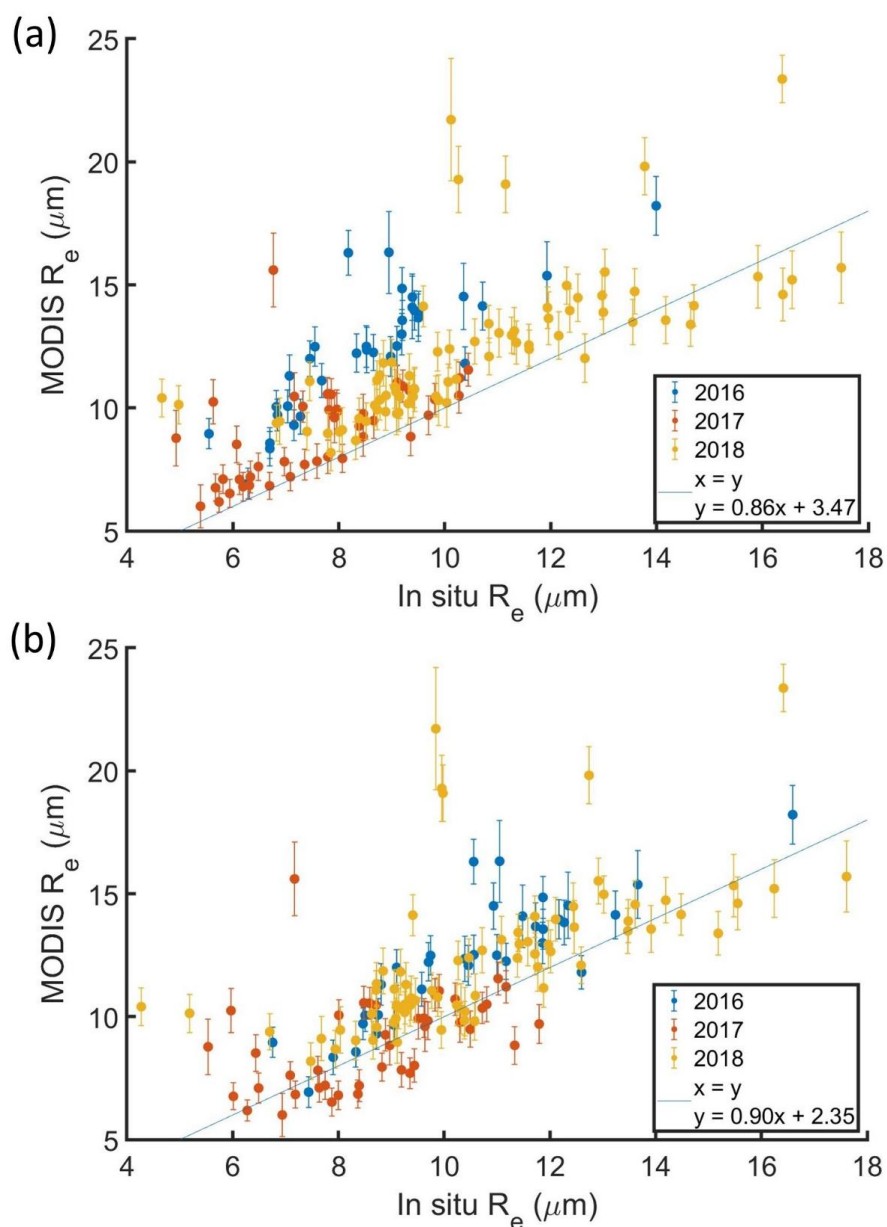

Figure A1: Same as Fig. 3a with in situ $R_e$ calculated (a) unscaled CAS and CDP $n(D)$ and (b) CAS
and CDP $n(D)$ scaled based on King LWC.



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
