# Peer review of "In Situ and Satellite-based Estimates of Cloud Properties and Aerosol-Cloud Interactions over the Southeast Atlantic Ocean"

_Atmospheric Chemistry and Physics, 2022_

## Referee Comment (RC2)

**Review: "In Situ and Satellite-based Estimates of Cloud Properties and Aerosol-Cloud Interactions over the Southeast Atlantic Ocean"**

The study uses aircraft *in situ* cloud property measurements to evaluate MODIS retrieved cloud properties over the Southeast Atlantic Ocean. Compared with previous efforts of evaluating MODIS cloud retrievals, this study further evaluates MODIS cloud retrievals under different ACI conditions in terms of aerosol 'contact' and 'separated' conditions relative to cloud top, which provides support for utilizing MODIS retrievals for ACI study over larger domains. Therefore, the manuscript fits well within the scope of Atmospheric Chemistry and Physics. Overall, the manuscript is well written and well organized. The manuscript is publishable in ACP. However, I do have some comments and concerns as listed below.

**Major comments:**

1. This study compares *in situ* and MODIS-based estimates of cloud properties in terms of cloud effective radius ($R_e$), cloud optical depth ($\tau$), and cloud droplet number concentration ($N_c$). As mentioned in the abstract, there are many previous studies of similar comparisons. Therefore, it is essential for this paper to discuss the results presented in this study related to previous comparisons (e.g., in the discussion section). What are the causes of the overestimations of $R_e$ and $\tau$ from MODIS retrievals?

2. Line 59-65: The introduction emphasizes that ACI over the southeast Atlantic Ocean is mainly related to the overly of biomass burning aerosol and the aerosol semi-direct radiative effect, but in section 4 it shows that aerosol indirect effect of causing large $Nc$ was observed from the differences between contact and separated cloud profiles. Does this mean that biomass burning aerosol exerts both indirect and semi-direct effects on clouds?

3. Line 536-538: 'The retrieval uncertainty for MODIS $R_e$ provided the largest source of error in calculating MODIS $Nc$ but compensating uncertainties for $\tau$, k, $C_w$, and $\alpha$ resulted in good agreement.' Can this be generalized to all marine stratocumulus under different aerosol loadings? For example, Meyer et al., (2013) show that the presence of absorbing aerosols overlying marine boundary layer clouds cause biases for MODIS retrieved $R_e$ and $\tau$.

   Reference: Meyer, K., Platnick, S., Oreopoulos, L., and Lee, D. (2013), Estimating the direct radiative effect of absorbing aerosols overlying marine boundary layer clouds in the southeast Atlantic using MODIS and CALIOP, *J. Geophys. Res. Atmos.*, 118, 4801–4815, doi:10.1002/jgrd.50449.

**Minor comments:**

1. Line 29: 'low biases in MODIS retrievals of cloud properties', this is confusing. The manuscript shows high biases of MODIS retrieved cloud effective radius and optical depth.

2. Line 58: references are needed here.

3. Line 104: the sentence needs a better structure. Maybe separate it into two sentences.

4. Table 1: for each flight date, the flight duration was usually several hours, but the total contact and separated sample time was only several hundred seconds. How 'contact and separated sample time' was selected?

5. Equations: should all items behind '/' be at the denominator or just the variable or constant right behind it at the denominator? Apparently, the expression of equation (6) is not consistent with equation (7) in terms of the usage of '/'.

6. Line 283: how deep can MODIS $R_e$ retrievals penetrate in terms of optical depth?

7. Line 406: reference? The study shows that 'On average, the MODIS Re and $\tau$ (11.3 μm and 11.7) were 1.6 μm and 2.3 higher than the in situ Re and $\tau$'. Apparently, the differences are larger than the MODIS average retrieval uncertainty of 7.5 and 5%.

---

## Author Comment (AC1)

Author responses to comments from Anonymous Referee #1 on acp-2022-374

*Reviewer comments are in red and italicized,* author responses are in black, and "text added to the manuscript is within quotes."

*The paper is well written and I only have a few minor comments. The paper is thorough, which is appreciated as it documents the steps of the retrieval and the satellite comparison in a way that will be helpful to others.*

The reviewer's comments are appreciated and a response to each comment is provided below.

*Line 55: It seems like the authors are only discussing the first indirect effect and not adjustments. They could also cite work by others discussing aerosol cloud adjustments in the context of meteorological confounding variables and causal ambiguity (Gryspeerdt et al., 2019; McCoy et al., 2020). They may also wish to cite (Wood et al., 2012).*

Thank you for pointing this out. To address this comment, the text has been edited as follows:

"However, ACI are often masked by meteorological conditions (Mauger and Norris, 2007), cloud adjustments to increasing $N_a$ like invigoration (Douglas and L'Ecuyer, 2021) or non-linear Liquid Water Path (LWP) responses to changes in $N_c$ (Gryspeerdt et al., 2019), and the modulation of aerosol properties by clouds and precipitation (Wood et al., 2012). These confounding influences can be addressed to some extent by constraining meteorological variables that affect LWP and comparing clouds with similar LWP or low precipitation rates (e.g., McCoy et al., 2020; G22)."

*Line 146: What is the scale of the profile? Does this mean that on a research flight the max height at which cloud occurred and the min height were used? If the profile is too extensive it is not clear if this is a particularly good assumption and it is unclear why the median was not used. I may have understood what is being referred to and a small amount of additional information here might be helpful to readers.*

The average cloud thickness (cloud top height minus cloud base height) was about 201 m (Gupta et al., 2022). The text describes the max/min cloud heights during each vertical transect through cloud rather than the entire research flight. The thresholds for in-cloud measurements are listed to describe how cloud top and cloud base were defined for each cloud profile. Gupta et al. (2021) described the procedures followed to ensure any observations of cumulus clouds above or below the stratocumulus clouds were not included in the observations.

For clarity, line 146 is edited to "For each individual vertical transect through marine stratocumulus, cloud top height….". The following line is also added: "During the ORACLES deployments, the average H was about 201 m (G22)."

*Line 343: this compensating uncertainty is consistent with earlier studies such as (Painemal & Zuidema, 2011) and (Grosvenor & Wood, 2014).*

Thank you for pointing this out. The following text was added: "as has been reported previously (PZ11; Grosvenor and Wood, 2014)."

*Line 402: Assuming uncorrelated random errors would tend to overestimate the error since earlier the authors showed that there were compensating errors?*

The compensation of errors is discussed in the context of calculating MODIS $N_c$ since the variables can be in the numerator or denominator in Eq. 7. However, that should be independent of any correlations or relationship between the errors outside of the MODIS $N_c$ calculation. It is unlikely the errors are correlated given the different sources and calculation procedures for the variables. For example, $C_w$, $k$, and $\alpha$ come from in situ data while $R_e$ and $\tau$ from MODIS retrievals.

*Line 431: Nit-picky, but since these distributions are non-normal (N is lognormal) the two sample t-test is not appropriate here.*

This is a fair point. The discussion of 95% confidence intervals is removed and the differences between the average values for the variables are listed. For further information, the reader is referred to previous studies (Gupta et al., 2021; 2022) where the average values were listed for each variable in addition to the 95% confidence intervals to allow the reader to compare the two parameters.

*Section 4: this section is interesting as it compares places where aerosols touch the cloud layer with places where they do not. The section is a bit excessively descriptive of the figures and could be shortened a bit. Rather than listing differences a table with (for instance) Mann-Whitney U-test statistics could be given. Giving values for differences between contact and non-contact is useful, but a bit hard to contextualize in that no information on the aerosol loading is given.*

Table 5 was added to replace some of the detailed discussion from the text. Section 4 was then shortened to avoid any repetition of the information already provided in the tables/figures.

Aerosol loading should not have a direct impact on the comparisons between in situ and MODIS estimates of cloud properties – the primary focus of the current study. For a comprehensive analysis of aerosol-cloud interactions as a function of aerosol loading, the reader is referred to Gupta et al. (2021, 2022) where the topic is discussed in significant detail with additional regimes defined based on both above-and below-cloud aerosols.

*Line 432: Why would SST, stability (either EIS or LTS- they are nearly identical in this region) be affecting N? It would only apply to tau and re I believe.*

SST or stability could indirectly affect $N_c$ through their impact on $R_e$ or precipitation. Meteorological variables were compared between contact and separated profiles to limit the confounding influence of local meteorology (Gupta et al., 2022). However, the differences were small or statistically insignificant. These parameters were chosen because they can exert an influence on LWP for marine stratocumulus and can influence the assessment of aerosol-cloud interactions as the reviewer comment above (and references therein) highlighted.

---

## Author Comment (AC2)

Author responses to comments from Anonymous Referee #2 on acp-2022-374

*Reviewer comments are in red and italicized,* author responses are in black, and "text added to the manuscript is within quotes."

*Review: "In Situ and Satellite-based Estimates of Cloud Properties and Aerosol-Cloud Interactions over the Southeast Atlantic Ocean"*

*The study uses aircraft in situ cloud property measurements to evaluate MODIS retrieved cloud properties over the Southeast Atlantic Ocean. Compared with previous efforts of evaluating MODIS cloud retrievals, this study further evaluates MODIS cloud retrievals under different ACI conditions in terms of aerosol 'contact' and 'separated' conditions relative to cloud top, which provides support for utilizing MODIS retrievals for ACI study over larger domains.*
*Therefore, the manuscript fits well within the scope of Atmospheric Chemistry and Physics. Overall, the manuscript is well written and well organized. The manuscript is publishable in ACP. However, I do have some comments and concerns as listed below.*

The reviewer's comments are appreciated and a response to each comment is provided below.

*Major comments:*

1. *This study compares in situ and MODIS-based estimates of cloud properties in terms of cloud effective radius ($R_e$), cloud optical depth ($\tau$), and cloud droplet number concentration ($N_c$). As mentioned in the abstract, there are many previous studies of similar comparisons. Therefore, it is essential for this paper to discuss the results presented in this study related to previous comparisons (e.g., in the discussion section). What are the causes of the overestimations of $R_e$ and $\tau$ from MODIS retrievals?*

The manuscript compares the MODIS biases and the factors likely to contribute to these biases with previous studies throughout the text. Comparisons of the MODIS biases are confined to previous studies of marine stratocumulus under similar conditions since the accuracy of MODIS retrievals can vary depending on factors such as cloud heterogeneity, solar and satellite viewing geometry, cloud thermodynamic phase, drizzle occurrence, etc. The reviewer is directed to the following text for comparisons between results presented here and existing literature:

- Lines 243 to 247: the MODIS $R_e$ biases were compared with previous studies.
- Lines 274 to 280: Factors affecting MODIS $R_e$ retrievals like the solar/sensor zenith angles and the retrieval wavelength dependence were compared with previous studies.
- Lines 332-333: The MODIS $N_c$ biases were compared with a previous study.
- Section 3.3.1: The values of parameters used to calculate MODIS $N_c$ are compared with previous studies.

Following the reviewer's comment regarding comparisons of the MODIS biases with previous studies, the following text was added to the discussion section:

"The positive biases in MODIS retrievals of cloud properties for marine stratocumulus over the southeast Atlantic were about 16 % for $R_e$, 30 % for $\tau$, and negligible for $N_c$, on average. However, the biases were within the overall uncertainty (in situ + MODIS) associated with the data. In comparison, previous studies have reported MODIS biases for Re and $\tau$ between 15 to 20 % (PZ11), 17 to 24 % (Min et al., 2012), and 20 to 40 % (Noble and Hudson, 2015), and negligible MODIS biases for Nc (PZ11, Braun et al., 2018; Gryspeerdt et al., 2022). Satellite estimates of $N_c$ and aerosol perturbations of $N_c$ over the southeast Atlantic have biases within 10 % of the in situ estimates."

Our hypotheses for the overestimation of $R_e$ and $\tau$ from MODIS retrievals are added to the discussion section:

"It is hypothesized that these biases could be reduced by addressing the in situ measurement uncertainty for k, the in situ derived uncertainty for $\alpha$ (e.g., Min et al., 2012; Merk et al., 2016; Braun et al., 2018; Witte et al., 2018), and the MODIS retrieval uncertainties associated with the bi-spectral retrieval technique (e.g., Fu et al., 2019; 2022)."

> *2.    Line 59-65: The introduction emphasizes that ACI over the southeast Atlantic Ocean is mainly related to the overly of biomass burning aerosol and the aerosol semi-direct radiative effect, but in section 4 it shows that aerosol indirect effect of causing large Nc was observed from the differences between contact and separated cloud profiles. Does this mean that biomass burning aerosol exerts both indirect and semi-direct effects on clouds?*

The indirect effect was introduced in the section but a link to the southeast Atlantic was not provided explicitly. Thus, leading to some confusion regarding the indirect effect on clouds in this domain. The following text was added to the introduction to outline the impact of the indirect effect on the marine stratocumulus over the southeast Atlantic Ocean based on previous work:

"The changes in $N_c$ and $R_e$ can lead to adjustments in precipitation formation processes and lead to higher cloud lifetime (Albrecht, 1989). An increase in $\tau$ and a decrease in precipitation rate associated with these ACI was reported for marine stratocumulus clouds over the southeast Atlantic Ocean (Gupta et al., 2021, hereafter G21; Gupta et al., 2022, hereafter G22)."

> *3.  Line 536-538: 'The retrieval uncertainty for MODIS $R_e$ provided the largest source of error in calculating MODIS $N_c$ but compensating uncertainties for $\tau$, k, $C_w$, and $\alpha$ resulted in good agreement.' Can this be generalized to all marine stratocumulus under different aerosol loadings? For example, Meyer et al., (2013) show that the*

*presence of absorbing aerosols overlying marine boundary layer clouds cause biases for MODIS retrieved $R_e$ and $\tau$.*

The part of the statement "*the retrieval uncertainty for MODIS $R_e$ provided the largest source of error in calculating MODIS $N_c$*" is primarily based on the equation used to calculate MODIS $N_c$. Given the exponential factor of 5/2 for $R_e$, the impact of the retrieval uncertainty for $R_e$ is amplified compared to other variables. This should not vary based on the aerosol loading given the impact of the overlying aerosols on the MODIS retrieval of $R_e$ is about 2 % for the southeast Atlantic (Meyer et al., 2015). As described in the discussion section, if the factors affecting MODIS retrievals are accounted for, this statement can be generalized to marine stratocumulus in other regions. Specifically, it is hypothesized the biases should be similar for homogeneous, warm, closed cell marine stratocumulus with low precipitation rates.

Reference: Meyer, K., Platnick, S., and Zhang, Z.: Simultaneously inferring above-cloud absorbing aerosol optical thickness and underlying liquid phase cloud optical and microphysical properties using MODIS, J. Geophys. Res.-Atmos., 120, 5524–5547, doi:10.1002/2015JD023128, 2015.

*Reference: Meyer, K., Platnick, S., Oreopoulos, L., and Lee, D. (2013), Estimating the direct radiative effect of absorbing aerosols overlying marine boundary layer clouds in the southeast Atlantic using MODIS and CALIOP, J. Geophys. Res. Atmos., 118, 4801– 4815, doi:10.1002/jgrd.50449.*

*Minor comments:*

1. *Line 29: 'low biases in MODIS retrievals of cloud properties', this is confusing. The manuscript shows high biases of MODIS retrieved cloud effective radius and optical depth.*

This was meant to convey the magnitude of the biases was small instead of the values being lower. The sentence is changed to "small biases in MODIS retrievals of cloud properties relative to in situ measurements" to avoid confusion.

2. *Line 58: references are needed here.*

A reference to Painemal and Zuidema (2011) is added here to refer to marine stratocumulus since that is the primary focus of the study. The following text was added at the end of the sentence: "such as marine stratocumulus clouds (Painemal and Zuidema, 2011, hereafter PZ11)."

3. *Line 104: the sentence needs a better structure. Maybe separate it into two sentences.*

The sentence was changed to "A number of studies have compared MODIS retrievals of marine stratocumulus cloud properties with in situ observations (PZ11; Min et al., 2012; Noble and Hudson, 2015; Braun et al., 2018; Witte et al., 2018). This study expands upon the existing

literature by using a larger in situ dataset which provides cloud and aerosol measurements under conditions of variable vertical separation between the aerosol and cloud layers."

4. *Table 1: for each flight date, the flight duration was usually several hours, but the total contact and separated sample time was only several hundred seconds. How 'contact and separated sample time' was selected?*

The "Time (UTC)" parameter in Table 1 does not refer to flight duration. Instead, it refers to the time range when cloud profiles were conducted. For example, on 06 Sep 2016, the first profile was started at 09.36 UTC and the last profile was concluded at 12:35 UTC. However, the aircraft was not always within cloud between these two timestamps. This is clarified within the text after Line 208: "The time range between the first and final cloud profile during each flight is listed in Table 1."

A typical flight was 7-9 hours long including cloud sampling for 1-2 hours, on average, because flight legs were also dedicated for aerosol sampling or for radiation retrievals. The cloud sampling time was determined using thresholds of $N_c$ and King LWC as defined in Section 2.

5. *Equations: should all items behind '/' be at the denominator or just the variable or constant right behind it at the denominator? Apparently, the expression of equation (6) is not consistent with equation (7) in terms of the usage of '/'.*

The formatting of the equations is changed to have stacked fractions and remove ambiguity.

6. *Line 283: how deep can MODIS $R_e$ retrievals penetrate in terms of optical depth?*

Line 179 in the manuscript states: "*Re16, Re21,* and *Re37* represent *Re* at 2 to 4 optical depths below cloud top depending on liquid water absorption and a weighting function based on vertical penetration of photons into cloud (McFarquhar and Heymsfield, 1998; Platnick, 2000)."

Thus, line 283 was changed to "Since each MODIS $R_e$ retrieval penetrates at least 2 optical depths into cloud…"

7. *Line 406: reference? The study shows that 'On average, the MODIS Re and $\tau$ (11.3 $\mu$m and 11.7) were 1.6 $\mu$m and 2.3 higher than the in situ Re and $\tau$'. Apparently, the differences are larger than the MODIS average retrieval uncertainty of 7.5 and 5%.*

No reference is provided since these values were calculated by averaging the uncertainty variable provided within the MODIS C6 product. The text is edited to clarity this as follows: "For MODIS $R_e$ and $\tau$, the error was defined as the average of the retrieval uncertainty provided within the MODIS C6 product (7.5 and 5 %, respectively)."

The differences are greater than the MODIS average retrieval uncertainty, but the MODIS biases are within the overall uncertainty after the in situ measurement uncertainty is added.

---

## Author Response (AR2)

Respected Editor,

The images were likely blurred while converting the Word document to PDF format. We have increased the quality of this conversion as possible with Word. The figures attached separately are provided at 300 dpi resolution.

Thanks.